# AugCal: Improving Sim2Real Adaptation by Uncertainty Calibration on Augmented Synthetic Images

**Prithvijit Chattopadhyay**[*]    **Bharat Goyal**    **Boglarka Ecsedi**    **Viraj Prabhu**    **Judy Hoffman**

Georgia Tech

{prithvijit3,bharatgoyal,becsedi3,virajp,judy}@gatech.edu

## Abstract

Synthetic data (Sim) drawn from simulators have emerged as a popular alternative for training models where acquiring annotated real-world images is difficult. However, transferring models trained on synthetic images to real-world applications can be challenging due to appearance disparities. A commonly employed solution to counter this Sim2Real gap is unsupervised domain adaptation, where models are trained using labeled Sim data and unlabeled Real data. Mispredictions made by such Sim2Real adapted models are often associated with miscalibration – stemming from overconfident predictions on real data. In this paper, we introduce AugCal, a simple training-time patch for unsupervised adaptation that improves Sim2Real adapted models by – (1) reducing overall miscalibration, (2) reducing overconfidence in incorrect predictions and (3) improving confidence score reliability by better guiding misclassification detection – all while retaining or improving Sim2Real performance. Given a base Sim2Real adaptation algorithm, at training time, AugCal involves replacing vanilla Sim images with strongly augmented views (Aug intervention) and additionally optimizing for a training time calibration loss on augmented Sim predictions (Cal intervention). We motivate AugCal using a brief analytical justification of how to reduce miscalibration on unlabeled Real data. Through our experiments, we empirically show the efficacy of AugCal across multiple adaptation methods, backbones, tasks and shifts.

## 1 Introduction

Most effective models for computer vision tasks (classification, segmentation, *etc.*) need to learn from a large amount of exemplar data (Dosovitskiy et al., 2020; Radford et al., 2021; Kirillov et al., 2023; Pinto et al., 2008) that captures real-world natural variations which may occur at deployment time. However, collecting and annotating such diverse real-world data can be prohibitively expensive – for instance, densely annotating a frame of Cityscapes (Cordts et al., 2016) can take upto $\sim 1.5$ hours! Machine-labeled synthetic images generated from off-the-shelf simulators can substantially reduce this need for manual annotation and physical data collection (Sankaranarayanan et al., 2018; Ros et al., 2016; Blaga & Nedevschi, 2019; Savva et al., 2019; Deitke et al., 2020; Chattopadhyay et al., 2021). Nonetheless, models trained on Sim data often exhibit subpar performance on Real data, primarily due to appearance discrepancies, commonly referred to as the Sim2Real gap. For instance, on GTAV (Sim) $\rightarrow$ Cityscapes (Real), an HRDA Sim-only model (Hoyer et al., 2022b) achieves an mIoU of only $53.01$, compared to $\sim 81$ mIoU attained by an equivalent model trained exclusively on Real data.

While there is significant effort in improving the realism of simulators (Savva et al., 2019; Richter et al., 2022), there is an equally large effort seeking to narrow this Sim2Real performance gap by designing algorithms that facilitate Sim2Real transfer. These methods encompass both *generalization* (Chattopadhyay* et al., 2023; Huang et al., 2021; Zhao et al., 2022). – aiming to ensure strong out-of-the-box Real performance of Sim trained models – and *adaptation* (Hoyer et al., 2022b;c; Vu et al., 2019; Rangwani et al., 2022) – attempting to adapt models using labeled Sim data and unlabeled Real data. Such generalization and adaptation methods have demonstrated notable success in reducing the Sim2Real performance gap. For instance, Pasta (Chattopadhyay* et al., 2023) (a *generalization* method) improves Sim2Real performance of a Sim-only model from

---

[*]Correspondence to PC

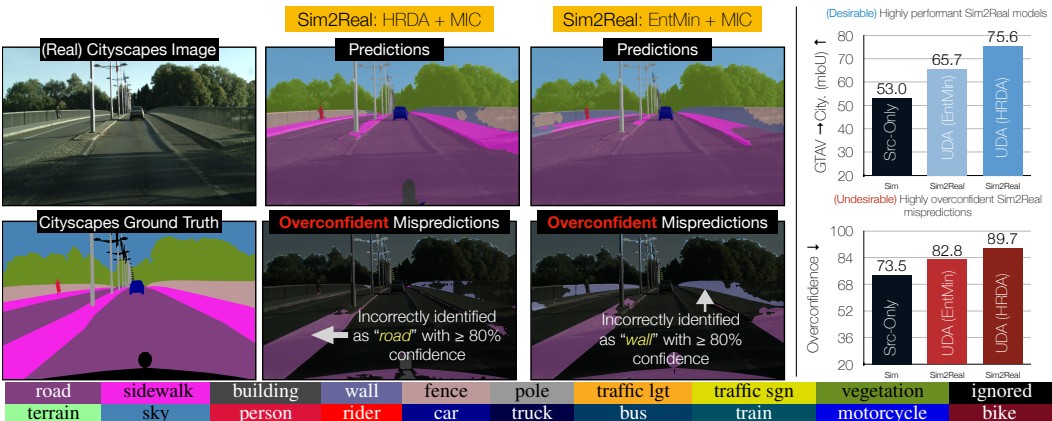

Figure 1: **Overconfident SIM2REAL mispredictions**. **[Left]** We show an example of what we mean by overconfident mispredictions. For SIM2REAL adaptation on GTAV→Cityscapes, we choose (DAFormer) HRDA + MIC (Hoyer et al., 2022c) and EntMin + MIC (Vu et al., 2019) (highly performant SIM2REAL methods) and show erroneous predictions on Cityscapes (bottom row). We can see that the model identifies *sidewalk* pixels as *road* (2nd column) and *fence* pixels as *wall* (3rd column) with very high confidence. **[Right]** We show how pervasive this "overconfidence" phenomena is. While better SIM2REAL adapted models – from (DAFormer) Source-Only (Hoyer et al., 2022b) to (DAFormer) EntMin + MIC (Vu et al., 2019) to (DAFormer) HRDA + MIC (Hoyer et al., 2022c) – exhibit improved transfer performance **[Top, Right]**, they also exhibit increased overconfidence in mispredictions **[Bottom, Right]**, affecting prediction reliability.

53.01 → 57.21 mIoU. Furthermore, HRDA + MIC (Hoyer et al., 2022c) (an *adaptation* approach) pushes performance even higher to 75.56 mIoU.

While SIM2REAL performance may increase both from generalization or adaptation methods, for safety-critical deployment scenarios, task performance is often not the sole factor of interest. It is additionally important to ensure SIM2REAL adapted models make *calibrated* and *reliable* predictions on REAL data. Optimal calibration on real data ensures that the model's confidence in its predictions aligns with the true likelihood of correctness. Deploying poorly calibrated models can have severe consequences, especially in high-stakes applications (such as autonomous driving), where users can place trust in (potentially) unreliable predictions (Tesla Crash, 2016; Michelmore et al., 2018). We find that mistakes made by SIM2REAL adaptation methods are often associated with miscalibration caused by *overconfidence* – highly confident incorrect predictions (see Fig. 1 Left). More interestingly, we find that as adaptation methods improve in terms SIM2REAL performance, the propensity to make overconfident mispredictions also increases (see Fig. 1 Right). Our focus in this paper is to devise training time solutions to mitigate this issue.

Calibrating deep neural networks (for such SIM2REAL adaptation methods) is crucial, as they routinely make overconfident predictions (Guo et al., 2017; Gawlikowski et al., 2021; Minderer et al., 2021). While various techniques address miscalibration on "labeled data splits" for in-distribution scenarios, maintaining calibration in the face of dataset shifts, like SIM2REAL, proves challenging due to lack of labeled examples in the target (REAL) domain. To address this, we propose AUGCAL, a training-time patch to ensure existing SIM2REAL adaptation methods make *accurate*, *calibrated* and *reliable* predictions on real data. When applied to a SIM2REAL adaptation framework, AUGCAL aims to satisfy three key criteria: (1) retain performance of the base SIM2REAL method, (2) reduce miscalibration and overconfidence and (3) ensure calibrated confidence scores translate to improved reliability. Additionally, to ensure broad applicability, AUGCAL aims to do so by making two minimally invasive changes to a SIM2REAL adaptation training pipeline. First, by AUGmenting (Cubuk et al., 2020; Chattopadhyay* et al., 2023) input SIM images during training using an AUG transform that reduces distributional distance between SIM and REAL images. Second, by additionally optimizing for a CALibration loss (Hebbalaguppe et al., 2022; Liang et al., 2020a; Liu et al., 2022) at training time on AUGmented SIM predictions. We devise AUGCAL based on an analytical rationale (see Sec. 3.2.1 and 3.2.2) illustrating how it helps reduce an upper bound on desired target (REAL) calibration error. Through our experiments on GTAV→Cityscapes and VisDA SIM2REAL, we demonstrate how AUGCAL helps reduce miscalibration on REAL data. To summarize, we make the following contributions:

- We propose AUGCAL, a training time patch, compatible with existing SIM2REAL adaptation methods that ensures SIM2REAL adapted models make *accurate* (measured via adaptation performance), *calibrated* (measured via calibration error) and *reliable* (measured via confidence guided misclassification detection) predictions.

- We conduct SIM2REAL adaptation experiments for object recognition (VisDA (Peng et al., 2017)) and semantic segmentation (GTAV (Sankaranarayanan et al., 2018)→Cityscapes (Cordts et al., 2016)) with three representative UDA methods (pseudo-label based self-training, entropy minimization and domain adversarial training) and show that applying AUGCAL– (1) improves or preserves adaptation performance, (2) reduces miscalibration and overconfidence and (3) improves the reliability of confidence scores.
- We show how AUGCAL improvements are effective across multiple backbones, AUG and CAL options and highlight choices that are more consistently effective across experimental settings.

## 2 RELATED WORK

**Unsupervised Domain Adaptation (UDA).** We focus on UDA algorithms to address *covariate shifts* in the SIM2REAL context (Chattopadhyay* et al., 2023; Choi et al., 2021; Zhao et al., 2022; Huang et al., 2021; Rangwani et al., 2022; Hoyer et al., 2022c; Sankaranarayanan et al., 2018; Ros et al., 2016). This involves adapting a model to an unseen target (REAL) domain using labeled samples from a source (SIM) domain and unlabeled samples from the target domain. Here, the source and target datasets share the same label space and labeling functions, but differences exist in the distribution of inputs (Farahani et al., 2021; Zhang et al., 2019). SIM2REAL UDA methods (Ganin & Lempitsky, 2014; Hoffman et al., 2018; Saenko et al., 2010; Tzeng et al., 2014) range from *feature distribution matching* (Ganin & Lempitsky, 2014; Long et al., 2018; Saito et al., 2018; Tzeng et al., 2017; Zhang et al., 2019), explicitly addressing *domain discrepancy* (Kang et al., 2019; Long et al., 2015; Tzeng et al., 2014; Rangwani et al., 2022), *entropy minimization* (Vu et al., 2019) or *pseudo-label guided self-training* (Hoyer et al., 2022b;a;c). We observe that existing SIM2REAL UDA methods usually improve performance at the expense of increasingly overconfident mispredictions on (REAL) target data (Wang et al., 2020b). Our proposed method, AUGCAL, is designed to retain SIM2REAL adaptation performance while reducing miscalibration on real data for existing methods. We conduct experiments on three representative UDA methods – Entropy Minimization (Vu et al., 2019), Self-training (Hoyer et al., 2022b) and Domain Adversarial Training (Rangwani et al., 2022).

**Confidence Calibration for Deep Networks.** For discriminative models, confidence calibration indicates the degree to which confidence scores associated with predictions align with the true likelihood of correctness (usually measured via ECE (Naeini et al., 2015)). Deep networks tend to be be very poor at providing calibrated confidence estimates (are overconfident) for their predictions (Guo et al., 2017; Gawlikowski et al., 2021; Minderer et al., 2021), which in turn leads to less reliable predictions for decision-making in safety-critical settings. Recent work (Guo et al., 2017) has also shown that calibration worsens for larger models and can decrease with increasing performance. Several works (Guo et al., 2017; Lakshminarayanan et al., 2017; Malinin & Gales, 2018) have explored this problem for modern architectures, and several solutions have also been proposed –including temperature scaling (prediction logits being divided by a scalar learned on a held-out set (Platt et al., 1999; Kull et al., 2017; Bohdal et al., 2021; Islam et al., 2021)) and trainable calibration objectives (training time loss functions that factor in calibration (Liang et al., 2020a; Karandikar et al., 2021)). Improving network calibration is even more challenging in out-of-distribution settings due to the simultaneous lack of ground truth labels and overconfidence on unseen samples (Wang et al., 2020b). Specifically, instead of methods that rely on temperature-scaling (Wang et al., 2020a; 2022) or maybe require an additional calibration split, AUGCAL explores the use of training time calibration objectives (Munir et al., 2022) to reduce micalibration for SIM2REAL shifts.

## 3 METHOD

### 3.1 BACKGROUND

**Notations.** Let $x$ denote input images and $y$ denote corresponding labels (from the label space $\mathcal{Y} = \{1, 2, ..., K\}$) drawn from a joint distribution $P(x, y)$. We focus on the classification case, where the goal is to learn a discriminative model $\mathcal{M}_\theta$ (with parameters $\theta$) that maps input images to the desired $K$ output labels, $\mathcal{M}_\theta : \mathcal{X} \rightarrow \mathcal{Y}$, using a softmax layer on top. The predictive probabilities for the given input can be expressed as $p_\theta(y|x) = \text{softmax}(\mathcal{M}_\theta(x))$. We use $\hat{y} = \arg\max_{y \in \mathcal{Y}} p_\theta(y|x)$ to denote the predicted label for $x$ and $c$ to denote the confidence in prediction.

**Unsupervised SIM2REAL Adaptation.** In unsupervised domain adaptation (UDA) for SIM2REAL settings, we assume access to a labeled (SIM) source dataset $D_S = \{(x_i^S, y_i^S)\}_{i=1}^{|S|}$ and an unlabeled (REAL) target dataset $D_T = \{x_i^T\}_{i=1}^{|T|}$. We assume $D_S$ and $D_T$ splits are drawn from source and target distributions $P^S(x, y)$ and $P^T(x, y)$ respectively. At training, we have access to $D = D_S \cup D_T$. We operate in the setting where source and target share the same label space, and discrepancies exist

only in input images. The model $\mathcal{M}_\theta$ is trained on labeled source images using cross entropy,

$$\sum_{i=1}^{|S|} \mathcal{L}_{CE}(x_i^S, y_i^S; \theta) = -\sum_{i=1}^{|S|} y_i^S \log p_\theta(\hat{y}_i^S | x_i^S) \text{ where } \hat{y}_i^S = \underset{y \in \mathcal{Y}}{\arg\max}\, p_\theta(y_i^S | x_i^S) \qquad (1)$$

UDA methods additionally optimize for an adaptation objective on labeled source and unlabeled target data ($\mathcal{L}_{UDA}$). The overall learning objective can be expressed as,

$$\min_\theta \underbrace{\sum_{i=1}^{|S|} \mathcal{L}_{CE}(x_i^S, y_i^S; \theta)}_{\text{Source Loss}} + \underbrace{\sum_{i=1}^{|T|} \sum_{j=1}^{|S|} \lambda_{UDA} \mathcal{L}_{UDA}(x_i^T, x_j^S, y_j^S; \theta)}_{\text{Source Target Adaptation Loss}} \qquad (2)$$

Different adaptation methods usually differ in terms of specific instantiations of this objective. While AUGCAL is applicable to any SIM2REAL adaptation method in principle, we conduct experiments with three popular methods – Entropy Minimization (Vu et al., 2019), Pseudo-Label driven Self-training (Hoyer et al., 2022b) (for semantic segmentation) and Domain Adversarial Training (Rangwani et al., 2022) (for object recognition). We provide more details on these methods in Sec. A.9 of appendix.

**Uncertainty Calibration.** For a perfectly calibrated classifier, the confidence in predictions should match the empirical frequency of correctness. Empirically, calibration can be measured using Expected Calibration Error (ECE) (Naeini et al., 2015). To measure ECE on a test set $D = \{(x_i, y_i)\}_{i=1}^{|D|}$, we first partition the test data into $B$ bins, $D_b = \{(x, y) \mid r_{b-1} \le c < r_b\}$, using the confidence values $c$ such that $b \in \{1, ..., B\}$ and $0 = r_0 \le r_1 \le r_2 \le \cdots \le r_B = 1$. Then, ECE measures the absolute differences between accuracy and confidence across instances in every bin,

$$\text{ECE} = \sum_{j=1}^{B} \frac{B}{|D|} \left| \frac{1}{B} \sum_{i \in D_j} \mathbf{1}_{(y_i = \hat{y}_i)} - \sum_{i \in D_j} c_i \right| \qquad (3)$$

We are interested in models that exhibit high-performance and low calibration error (ECE). Note that Eqn 3 alone does not indicate if a model is overconfident. We define overconfidence (OC) as the expected confidence on mispredictions. Prior work on improving calibration in out-of-distribution (OOD) settings (Wang et al., 2022) and domain adaptation scenarios (Wang et al., 2020b) typically rely on techniques like temperature scaling. These methods often necessitate additional steps, such as employing a separate calibration split or domains (Gong et al., 2021) or training extra models (e.g., logistic discriminators for source and target features (Wang et al., 2020b)). In contrast, we consider using training time calibration objectives (Liang et al., 2020b; Hebbalaguppe et al., 2022; Liu et al., 2022) that can be optimized in addition to task-specific objectives for improved calibration.

## 3.2 AUGCAL

### 3.2.1 REDUCING MISCALIBRATION ON (REAL) TARGET

Recall that $P^S(x, y)$ and $P^T(x, y)$ denote the source (SIM) and target (REAL) data distributions. We assume $P(x, y)$ factorizes as $P(x, y) = P(x)P(y|x)$. We assume covariate shift conditions between $P^S$ and $P^T$, *i.e.*, $P^T(x) \ne P^S(x)$ while $P^T(y|x) = P^S(y|x)$ – discrepancies across distributions exist only in input images. When training a model, we can only draw "labeled samples" $(x, y)$ from $P^S(x, y)$. We do not have access to labels from $P^T(x, y)$. Our goal is to reduce miscalibration on (unlabeled) target data using training time calibration losses. Let $\mathcal{L}_{\text{CAL}}(x, y)$ denote such a calibration loss we can minimize (on labeled data). Using importance sampling (Cortes et al., 2010), we can get an estimate of the desired calibration loss on target data as,

$$\underset{x,y \sim P^T(x,y)}{\mathbb{E}} [\mathcal{L}_{\text{CAL}}(x, y)] = \int_x \int_y \mathcal{L}_{\text{CAL}}(x, y) P^T(x, y)\, dx\, dy$$

$$= \int_x \int_y \mathcal{L}_{\text{CAL}}(x, y) \frac{P^T(x) P^T(y|x)}{P^S(x) P^S(y|x)} P^S(x, y)\, dx\, dy$$

$$= \underset{x,y \sim P^S(x,y)}{\mathbb{E}} \left[ \underbrace{w_S(x)}_{\text{Importance Weight}} \underbrace{\mathcal{L}_{\text{CAL}}(x, y)}_{\text{Source Loss}} \right] \qquad (4)$$

where $w_S(x) = \frac{P^T(x)}{P^S(x)}$ denotes the importance weight. Assuming $\mathcal{L}_{\text{CAL}}(x,y) \geq 0$[1], we can obtain an upper bound on step 4 (Pampari & Ermon, 2020; Wang et al., 2020b) as

$$\mathop{\mathbb{E}}_{x,y \sim P^T(x,y)}\left[\mathcal{L}_{\text{CAL}}(x,y)\right] = \mathop{\mathbb{E}}_{x,y \sim P^S(x,y)}\left[w_S(x)\mathcal{L}_{\text{CAL}}(x,y)\right] \tag{5}$$

$$\leq \sqrt{\mathop{\mathbb{E}}_{P^S(x)}\left[w_S(x)^2\right] \mathop{\mathbb{E}}_{P^S(x,y)}\left[\mathcal{L}_{\text{CAL}}(x,y)^2\right]} \tag{6}$$

$$\leq \frac{1}{2}\Big( \underbrace{\mathop{\mathbb{E}}_{P^S(x)}\left[w_S(x)^2\right]}_{\text{Shift Dependent}} + \underbrace{\mathop{\mathbb{E}}_{P^S(x,y)}\left[\mathcal{L}_{\text{CAL}}(x,y)^2\right]}_{\text{Source Dependent}} \Big) \tag{7}$$

where steps 6 and 7 use the Cauchy-Schwarz and AM-GM inequalities respectively. For a given model, the second RHS term in inequation 7 is computed purely on labeled samples from the source distribution and can therefore be optimized to convergence over the course of training. The gap in $\mathcal{L}_{\text{CAL}}$ across source and target is dominated by the importance weight (first term). Following (Cortes et al., 2010), the first term can also be expressed as,

$$\mathop{\mathbb{E}}_{P^S}\left[w_S(x)^2\right] = d_2\big(P^T(x)||P^S(x)\big) \tag{8}$$

where $d_\alpha(P||Q) = \left[\sum_x \frac{P^\alpha(x)}{Q^{\alpha-1}(x)}\right]^{\frac{1}{\alpha-1}}$ with $\alpha > 0$ is the exponential in base 2 of the Renyi-divergence (Rényi, 1960) between distributions $P$ and $Q$. The calibration error gap between source and target distributions is therefore, dominated by the divergence between source and target distributions. Consequently, inequation 7 can be expressed as,

$$\underbrace{\mathop{\mathbb{E}}_{x,y \sim P^T(x,y)}\left[\mathcal{L}_{\text{CAL}}(x,y)\right]}_{\text{Target Calibration Loss}} \leq \underbrace{\frac{1}{2}d_2\big(P^T(x)||P^S(x)\big)}_{\text{Source and Target Divergence}} + \underbrace{\frac{1}{2}\mathop{\mathbb{E}}_{P^S(x,y)}\left[\mathcal{L}_{\text{CAL}}(x,y)^2\right]}_{\text{Source Calibration Loss}} = \underbrace{U(S,T)}_{\text{Upper Bound}} \tag{9}$$

where $U(S,T)$ denotes the upper bound on target calibration loss. Therefore, to effectively reduce miscalibration on target data, one needs to reduce the upper bound, $U(S,T)$, which translates to (1) reducing miscalibration on source data (second red term in 9) and (2) reducing the distributional distance between input distributions across source and target (first blue term in 9).

### 3.2.2 WHY AUGCAL?

Based on the previous discussion, to improve calibration on target data, one can always invoke a training time calibration intervention (CAL) on labeled source data to reduce $\mathbb{E}_{P^S(x,y)}[\mathcal{L}_{\text{CAL}}(x,y)]$. In practice, after training, we can safely assume that $\mathbb{E}_{P^S(x,y)}[\mathcal{L}_{\text{CAL}}(x,y)] = \epsilon \to 0$ (for some very small $\epsilon$). We note that while this is useful and necessary, it is not sufficient. This is precisely where we make our contribution. To reduce both (red and blue) terms in 9, we introduce AUGCAL. To do this, in addition to a training time calibration loss, $\mathcal{L}_{\text{CAL}}$, AUGCAL assumes that access to an additional AUG transformation that satisfies the following properties:

1. $d_2\big(P^T(x)||P^S(\text{AUG}(x))\big) \leq d_2\big(P^T(x)||P^S(x)\big)$

2. After training, $\mathbb{E}_{P^S(x,y)}\left[\mathcal{L}_{\text{CAL}}(x,y)\right] \approx \mathbb{E}_{P^S(x,y)}\left[\mathcal{L}_{\text{CAL}}(\text{AUG}(x),y)\right] = \epsilon \to 0$

Property 1 states that the chosen AUG transformation brings transformed source data closer to target (or reduces SIM2REAL distributional distance). Property 2 states that over the course of training, irrespective of the data $\mathcal{L}_{\text{CAL}}(x,y)$ is optimized on (AUG transformed or clean source), $\mathbb{E}_{P^S(x,y)}\left[\mathcal{L}_{\text{CAL}}(\cdot,\cdot)\right]$ can achieve a sufficiently small value close to 0. Given an AUG transformation that satisfies the above stated properties, we can claim,

$$U^{\text{AUG}}(S,T) \leq U(S,T) \tag{10}$$

where

$$U^{\text{AUG}}(S,T) = \frac{1}{2}d_2\big(P^T(x)||P^S(\text{AUG}(x))\big) + \frac{1}{2}\mathop{\mathbb{E}}_{P^S(x,y)}\left[\mathcal{L}_{\text{CAL}}(\text{AUG}(x),y)^2\right] \tag{11}$$

---

[1] We make the reasonable assumption that the calibration loss function is always non-negative.

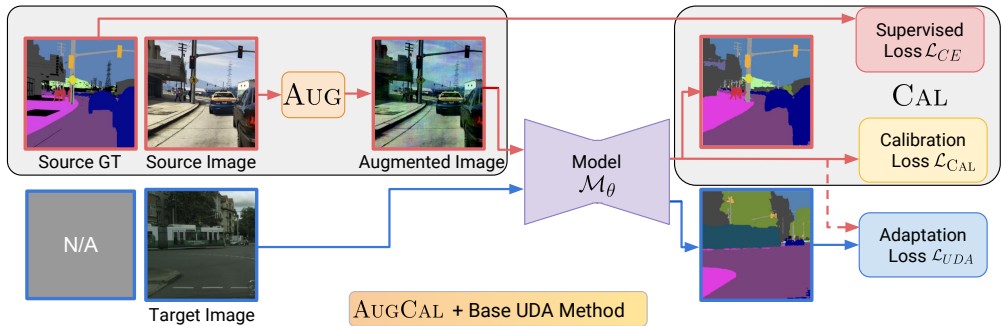

Figure 2: **AUGCAL pipeline**. AUGCAL consists of two key interventions on an existing SIM2REAL adaptation method. First source SIM images are augmented via an AUG transform. Supervised losses for SIM images are computed on the augmented image predictions. Additionally, AUGCAL optimizes for a calibration loss on AUGmented SIM predictions.

That is, an appropriate AUG transform, when coupled with $\mathcal{L}_{\text{CAL}}$, helps reduce a tighter upper bound on the target calibration error than CAL. We call this intervention – coupling AUG and CAL– AUGCAL. Naturally, the effectiveness of AUGCAL is directly dependent on the choice of AUG that satisfies the aforementioned properties.

Table 1: **Eligible AUG choices.** AUG transformations that reduce SIM2REAL distance and satisfy Property-1.

| SIM2REAL | 100x (RBF) MMD | | |
| | (SIM, REAL) | (PASTA-SIM, REAL) | (R.Aug-SIM, REAL) |
|---|---|---|---|
| VisDA | 4.806 | **4.171** | **3.464** |
| GTAV→City. | 5.795 | **5.214** | **5.213** |

While most augmentations can satisfy property 2, to check if an augmentation is valid according to property 1, we compute RBF Kernel based MMD distances for (SIM, REAL) and (AUGmented SIM, REAL) feature pairs using a trained model.[2] In Table. 1, we show how PASTA (Chattopadhyay* et al., 2023) and RandAugment (Cubuk et al., 2020), two augmentations effective for SIM2REAL transfer, satisfy these considerations on multiple shifts (read Table. 1 left to right). PASTA and RandAug are additionally (1) inexpensive when combined with SIM2REAL UDA methods, and (2) generally beneficial for SIM2REAL shifts (PASTA via SIM2REAL specific design and RandAug via chained photometric operations).

### 3.2.3 AUGCAL INSTANTIATION

Given a SIM2REAL adaptation method, AUGCAL additionally optimizes for improved calibration on augmented SIM source images. Since AUGCAL is applicable to any existing SIM2REAL adaptation method, we abstract away the adaptation component associated with the pipeline and denote as $\mathcal{L}_{UDA}$ (see Eqn. 15). The steps involved in AUGCAL are illustrated in Fig. 2. Given a mini-batch, we first generate augmented views, AUG($x^S$), for SIM images $x^S$. Then, during training, we optimize $\mathcal{L}_{CE}$ on those augmented SIM views and $\mathcal{L}_{UDA}$ for adaptation. To improve calibration under augmentations, we optimize an additional $\mathcal{L}_{\text{CAL}}$ loss on augmented SIM images. The overall AUGCAL optimization problem can be expressed as,

$$\min_\theta \underbrace{\sum_{i=1}^{|S|} \mathcal{L}_{CE}(\text{AUG}(x_i^S), y_i^S; \theta)}_{\text{Source Task Loss}} + \underbrace{\sum_{i=1}^{|T|}\sum_{j=1}^{|S|} \lambda_{UDA}\mathcal{L}_{UDA}(x_i^T, \text{AUG}(x_j^S), y_j^S; \theta)}_{\text{Source Target Adaptation Loss}} + \underbrace{\sum_{i=1}^{|S|} \lambda_{\text{CAL}}\mathcal{L}_{\text{CAL}}(\text{AUG}(x_i^S), y_i^S; \theta)}_{\text{Source Calibration Loss}}$$

(12)

where $\lambda_{UDA}$ and $\lambda_{\text{CAL}}$ denote the respective loss coefficients and the changes to a vanilla SIM2REAL adaptation framework are denoted in teal.

**Choice of "AUG".** Strongly augmenting SIM images during training has proven useful for SIM2REAL transfer (Chattopadhyay* et al., 2023; Huang et al., 2021; Zhao et al., 2022; Cubuk et al., 2020). For AUGCAL, we are interested in augmentations that satisfy the properties outlined in Sec. 3.2.2. As stated earlier, we find that RandAugment (Cubuk et al., 2020) and PASTA (Chattopadhyay* et al., 2023) empirically satisfy this criteria. We use both of them for our experiments and provide details on the operations for these AUG transforms in Sec. A.4 of appendix.

**Choice of "CAL" ($\mathcal{L}_{\text{CAL}}$).** AUGCAL relies on using training time calibration losses to reduce miscalibration. Prior work in uncertainty calibration has considered several auxiliary objectives to

---

[2]Under bounded importance weight assumptions, MMD can be interpreted as an upper bound on KL divergence (Wang & Tay, 2022).

calibrate a model being trained to reduce negative log-likelihood (NLL) (Hebbalaguppe et al., 2022; Kumar et al., 2018; Karandikar et al., 2021; Liang et al., 2020a). For "CAL" in AUGCAL, while we consider multiple calibration losses – DCA (Liang et al., 2020b), MbLS (Liu et al., 2022) and MDCA (Hebbalaguppe et al., 2022) – and find that DCA, simple "difference between confidence and accuracy (DCA)" loss proposed in (Liang et al., 2020a) is more consistently effective across experimental settings. DCA can be expressed as,

$$\mathcal{L}_{\text{CAL}} = \frac{1}{|S|} \left| \sum_{i=1}^{|S|} \mathbf{1}_{(y_i^S = \hat{y}_i^S)} - \sum_{i=1}^{|S|} p_\theta(\hat{y}_i^S | x_i^S) \right| \tag{13}$$

where $\mathbf{1}_{(y_i^S = \hat{y}_i^S)}$ and $p_\theta(\hat{y}_i^S | x_i^S)$ denote the correctness and confidence scores associated with predictions. The DCA loss forces the mean predicted confidence over training samples to match accuracy. In the following sections, we empirically validate AUGCAL across adaptation methods.

## 4 EXPERIMENTAL DETAILS

We conduct SIM2REAL adaptation experiments across two tasks – Semantic Segmentation (SemSeg) and Object Recognition (ObjRec). For our experiments, we train models using labeled SIM images and unlabeled REAL images. We test trained models on REAL images.

**SIM2REAL Shifts.** For SemSeg, we conduct experiments on the GTAV→Cityscapes shift. GTAV (Sankaranarayanan et al., 2018) consists of $\sim$ 25k densely annotated SIM ground-view images and Cityscapes (Cordts et al., 2016) consists of $\sim$ 5k REAL ground view images. We report all metrics on the Cityscapes validation split. For ObjRec, we conduct experiments on the VisDA SIM2REAL benchmark. VisDA (Peng et al., 2017) consists of $\sim$ 152k SIM images and $\sim$ 55k REAL images across 12 classes. We report all metrics on the validation split of (REAL) target images.

**Models.** We check AUGCAL compatibility with both CNN and Transformer based architectures. For SemSeg, we consider DeepLabv2 (Chen et al., 2017) (with a ResNet-101 (He et al., 2016) backbone) and DAFormer (Hoyer et al., 2022a) (with an MiT-B5 (Xie et al., 2021) backbone) architectures. For ObjRec, we consider ResNet-101 and ViT-B/16 (Dosovitskiy et al., 2020) backbones with bottleneck layers as classifiers. We start with backbones pre-trained on ImageNet (Deng et al., 2009).

**Adaptation Methods.** We consider three representative SIM2REAL adaptation methods for our experiments. For SemSeg, we consider entropy minimization (EntMin) (Vu et al., 2019) and high-resolution domain adaptive semantic segmentation (HRDA) (Hoyer et al., 2022b). For ObjRec, we consider smooth domain adversarial training (SDAT) (Rangwani et al., 2022). For both tasks, we further improve performance with masked image consistency (MIC) (Hoyer et al., 2022c) on target images during training. We use MIC (Hoyer et al., 2022c)'s implementations of the adaptation algorithms and provide more training details in Sec. A.5 of appendix.

**Calibration Metrics.** We use ECE to report overall confidence calibration on REAL images. Since we are interested in reducing overconfident mispredictions, we also report calibration error on incorrect samples (IC-ECE) (Wang et al., 2022) and mean overconfidence for mispredictions (OC).

**Reliability Metrics.** While reducing overconfidence and improving calibration on real data is desirable, this is a proxy for the true goal of improving model reliability. To assess reliability, following prior work (de Jorge et al., 2023; Malinin et al., 2019), we measure whether calibrated confidence scores can better guide misclassification detection. To measure this, we use Prediction Rejection Ratio (PRR) (Malinin et al., 2019), which if high (positive and close to 100) indicates that confidence scores can be used as reliable indicators of performance (details in Sec. A.6 of appendix).

Unless specified otherwise, we use PASTA as the choice of AUG and DCA (Liang et al., 2020b) as the choice of CAL in AUGCAL. We use $\lambda_{\text{CAL}} = 1$ for DCA.

## 5 FINDINGS

### 5.1 IMPROVING SIM2REAL ADAPTATION

Recall that when applied to a SIM2REAL adaptation method, we expect AUGCAL to – (1) retain SIM2REAL transfer performance, (2) reduce miscalibration and overconfidence and (3) ensure calibrated confidence scores translate to improved model reliability. We first verify these criteria.

▷ **AUGCAL improves or retains SIM2REAL adaptation performance.** Since AUGCAL intervenes on an existing SIM2REAL adaptation algorithm, we first verify that encouraging better calibration does not adversely impact SIM2REAL adapation performance. We find that performance is either retained or improved (*e.g.*, for EntMin + MIC in Table. 2 (a)) as miscalibration is reduced (Tables. 2(a) and (b), Perf. columns).

Table 2: **AUGCAL ensures SIM2REAL adapted models make accurate, calibrated and reliable predictions.** We find that applying AUGCAL to multiple SIM2REAL adaptation methods across tasks leads to better calibration (ECE, IC-ECE), reduced overconfidence (OC) and improved reliability (PRR) – all while retaining or improving transfer performance. Highlighted rows are AUGCAL variants of the base methods. For AUGCAL, we use PASTA as AUG and DCA as CAL. $\pm$ indicates standard error.

| Method | Perf. ($\uparrow$) mIoU | Calibration Error ($\downarrow$) ECE | IC-ECE | OC | Reliability ($\uparrow$) PRR |
|---|---|---|---|---|---|
| 1 EntMin + MIC | 65.71 | $5.34_{\pm0.35}$ | $77.73_{\pm0.26}$ | $82.83_{\pm0.55}$ | $45.93_{\pm0.54}$ |
| 2 + AUGCAL | **70.31** | $\mathbf{3.43}_{\pm0.29}$ | $\mathbf{72.97}_{\pm0.26}$ | $\mathbf{82.80}_{\pm0.57}$ | $\mathbf{62.66}_{\pm0.55}$ |
| 3 HRDA + MIC | 75.56 | $2.86_{\pm0.10}$ | $81.92_{\pm0.14}$ | $89.72_{\pm0.48}$ | $68.91_{\pm0.46}$ |
| 4 + AUGCAL | **75.90** | $\mathbf{2.45}_{\pm0.09}$ | $\mathbf{79.09}_{\pm0.16}$ | $\mathbf{88.26}_{\pm0.49}$ | $\mathbf{70.35}_{\pm0.51}$ |

(a) GTAV→Cityscapes. (DAFormer).

| Method | Perf. ($\uparrow$) mIoU | Calibration Error ($\downarrow$) ECE | IC-ECE | OC | Reliability ($\uparrow$) PRR |
|---|---|---|---|---|---|
| 1 SDAT + MIC | $92.53_{\pm0.28}$ | $7.67_{\pm0.49}$ | $91.45_{\pm0.63}$ | $89.13_{\pm1.29}$ | $63.78_{\pm2.12}$ |
| 2 + AUGCAL | $\mathbf{92.87}_{\pm0.06}$ | $\mathbf{6.84}_{\pm0.10}$ | $\mathbf{89.25}_{\pm0.36}$ | $\mathbf{85.74}_{\pm0.36}$ | $\mathbf{67.80}_{\pm0.78}$ |

(b) VisDA SIM2REAL. (ViT-B).

Table 3: **AUGCAL is better than applying AUG or CAL alone.** On GTAV→Cityscapes and VisDA, we show that AUGCAL improves over just augmented SIM training (AUG) or just optimizing for calibration on SIM data (CAL). For AUGCAL, we use PASTA as AUG and DCA as CAL. $\pm$ indicates standard error.

| Method | Perf. ($\uparrow$) mIoU | Calibration Error ($\downarrow$) ECE | IC-ECE | Reliability ($\uparrow$) PRR |
|---|---|---|---|---|
| 1 EntMin + MIC | 65.71 | $5.34_{\pm0.35}$ | $77.73_{\pm0.26}$ | $45.93_{\pm0.54}$ |
| 2 + AUG | 67.58 | $4.30_{\pm0.33}$ | $77.59_{\pm0.25}$ | $48.05_{\pm0.53}$ |
| 3 + CAL | 68.70 | $4.04_{\pm0.26}$ | $75.86_{\pm0.26}$ | $52.52_{\pm0.54}$ |
| 4 + AUGCAL | **70.31** | $\mathbf{3.43}_{\pm0.29}$ | $\mathbf{72.97}_{\pm0.26}$ | $\mathbf{62.66}_{\pm0.55}$ |

(a) GTAV→Cityscapes. (DAFormer).

| Method | Perf. ($\uparrow$) mAcc | Calibration Error ($\downarrow$) ECE | IC-ECE | Reliability ($\uparrow$) PRR |
|---|---|---|---|---|
| 1 SDAT + MIC | $92.53_{\pm0.28}$ | $7.67_{\pm0.49}$ | $91.45_{\pm0.63}$ | $63.78_{\pm2.12}$ |
| 2 + AUG | $92.69_{\pm0.15}$ | $9.48_{\pm1.99}$ | $90.48_{\pm0.33}$ | $65.68_{\pm0.58}$ |
| 3 + CAL | $91.63_{\pm0.71}$ | $7.30_{\pm0.09}$ | $91.19_{\pm0.13}$ | $66.62_{\pm1.70}$ |
| 4 + AUGCAL | $\mathbf{92.87}_{\pm0.06}$ | $\mathbf{6.84}_{\pm0.10}$ | $\mathbf{89.25}_{\pm0.36}$ | $\mathbf{67.80}_{\pm0.78}$ |

(b) VisDA SIM2REAL (ViT-B).

▷ **AUGCAL reduces miscalibration post SIM2REAL adaptation.** On both GTAV→Cityscapes and VisDA, we find that AUGCAL consistently reduces miscalibration of the base method by reducing overconfidence on incorrect predictions. This is evident in how AUGCAL variants of the base adaptation methods have lower ECE, IC-ECE and OC values (AUGCAL rows, Calibration columns in Tables. 2 (a) and (b)). As an example, to illustrate the effect of improved calibration on real data, in Fig. 3, we show how applying AUGCAL can improve the proportion of per-pixel SemSeg predictions that are accurate and have high-confidence ($> 0.95$).

▷ **AUGCAL improvements in calibration improve reliability.** As noted earlier, we additionally investigate the extent to which calibration improvements for SIM2REAL adaptation translate to reliable confidence scores – via misclassification detection on REAL target data (see Sec. 4), as measured by PRR. We find that AUGCAL consistently improves PRR of the base SIM2REAL adaptation method (PRR columns for AUGCAL rows in Tables. 2(a) and (b)) – ensuring that predictions made AUGCAL variants of a base model are more trustworthy.

### 5.2 ANALYZING AUGCAL

We now analyze different aspects of AUGCAL.

▷ **Applying AUGCAL is better than applying just AUG or CAL.** In Sec. 3.2.1 and 3.2.2, we discuss how AUGCAL can be more effective in reducing target miscalibration than just optimizing for improved calibration on labeled SIM images. We verify this empirically in Tables. 3(a) and (b) for SemSeg and ObjRec. We show that while AUG and CAL, when applied individually, improve over a base SIM2REAL method, they fall short of improvements offered by AUGCAL.

▷ **AUGCAL is applicable across multiple AUG choices.** In Sec. 3.2.2 and Table. 1, we show how both PASTA (Chattopadhyay* et al., 2023) and RandAugment (Cubuk et al., 2020) are eligible for AUGCAL. In Table. 4(a), we fix DCA as CAL and find that both PASTA and RandAugment are effective in retaining or improving performance, reducing miscalibration and improving reliability.

▷ **Ablating CAL choices for AUGCAL.** For completeness, we also conduct experiments by fixing PASTA as AUG and ablating the choice of CAL in AUGCAL. We consider recently proposed training time calibration objectives – Difference of Confidence and Accuracy (DCA) (Liang et al., 2020b), Multi-class Difference in Confidence and Accuracy (MDCA) (Hebbalaguppe et al., 2022) and Margin-based Label Smoothing (MbLS) (Liu et al., 2022) – as potential CAL choices (results for SemSeg outlined in Table. 4(b)). We find that while MDCA and MbLS can be helpful, DCA is more consistently helpful across tasks and settings.

▷ **AUGCAL is applicable across multiple task backbones.** Different architectures – CNNs and Transformers – are known to exhibit bias towards different properties in images (shape, texture, *etc.*) (Naseer et al., 2021). Since the choice of AUG transform (which can alter such properties) is central to the efficacy of AUGCAL, we verify if AUGCAL is effective across both CNN and Transformer backbones. To do this, we conduct our SemSeg, ObjRec experiments with both transformer (DAFormer, ViT-B) and CNN (DeepLabv2-R101, ResNet-101) architectures. We find that AUGCAL

Table 4: **Ablating AUG and CAL choices in AUGCAL.** For a DAFormer model on GTAV→Cityscapes, AUGCAL successfully reduces miscalibration and produces reliable confidence scores for SIM2REAL adaptation using both PASTA (P) and RandAug (R) as AUG choices. We also ablate the choice of CAL in AUGCAL across DCA, MDCA and MbLS and find that DCA is more consistently effective in reducing miscalibration across tasks and settings. $\lambda_{CAL} = 1$ for MDCA and $\lambda_{CAL} = 0.1, m = 10$ for MbLS. $\pm$ indicates standard error.

| Method | AUG | Perf. (↑) mIoU | Calibration Error (↓) ECE | IC-ECE | Reliability (↑) PRR |
|---|---|---|---|---|---|
| 3 EntMin | | 65.71 | $5.34_{\pm 0.35}$ | $77.73_{\pm 0.26}$ | $45.93_{\pm 0.54}$ |
| 4 + AUGCAL | P | 70.31 | $3.43_{\pm 0.29}$ | $72.97_{\pm 0.26}$ | $62.66_{\pm 0.55}$ |
| 4 + AUGCAL | R | 70.65 | $2.34_{\pm 0.14}$ | $73.77_{\pm 0.21}$ | $66.65_{\pm 0.46}$ |
| 7 HRDA | | 75.56 | $2.86_{\pm 0.10}$ | $81.92_{\pm 0.14}$ | $68.91_{\pm 0.46}$ |
| 8 + AUGCAL | P | 75.90 | $2.45_{\pm 0.09}$ | $79.09_{\pm 0.16}$ | $70.35_{\pm 0.51}$ |
| 8 + AUGCAL | R | 74.10 | $2.77_{\pm 0.17}$ | $77.94_{\pm 0.18}$ | $69.46_{\pm 0.46}$ |

(a) Ablating AUG in AUGCAL. (CAL = DCA).

| Method | CAL | Perf. (↑) mIoU | Calibration Error (↓) ECE | IC-ECE | Reliability (↑) PRR |
|---|---|---|---|---|---|
| 3 EntMin + MIC | | 65.71 | $5.34_{\pm 0.35}$ | $77.73_{\pm 0.26}$ | $45.93_{\pm 0.54}$ |
| 4 + AUGCAL | DCA | 70.31 | $3.43_{\pm 0.29}$ | $72.97_{\pm 0.26}$ | $62.66_{\pm 0.55}$ |
| 4 + AUGCAL | MDCA | 69.50 | $3.22_{\pm 0.26}$ | $72.65_{\pm 0.25}$ | $59.96_{\pm 0.51}$ |
| 4 + AUGCAL | MbLS | 68.77 | $2.90_{\pm 0.24}$ | $72.53_{\pm 0.23}$ | $61.57_{\pm 0.48}$ |
| 7 HRDA + MIC | | 75.56 | $2.86_{\pm 0.10}$ | $81.92_{\pm 0.14}$ | $68.91_{\pm 0.46}$ |
| 8 + AUGCAL | DCA | 75.90 | $2.45_{\pm 0.09}$ | $79.09_{\pm 0.16}$ | $70.35_{\pm 0.51}$ |
| 8 + AUGCAL | MDCA | 75.50 | $2.92_{\pm 0.15}$ | $80.76_{\pm 0.16}$ | $68.45_{\pm 0.47}$ |
| 8 + AUGCAL | MbLS | 71.15 | $2.68_{\pm 0.15}$ | $70.21_{\pm 0.42}$ | $69.33_{\pm 0.47}$ |

(b) Ablating CAL in AUGCAL. (AUG = PASTA)

Figure 3: **AUGCAL increases the proportion of "accurate" and "certain" predictions**. For a (DAFormer) HRDA + MIC (row 1) and EntMin + MIC (row 2) on GTAV→Cityscapes, we show how different interventions affect the proportion of "accurate" and "certain" (confidence $> 0.95$) predictions (indicated in gray per column). Regions in black do not satisfy the "accurate" and "certain" filtering criteria. We see that compared to a base adaptation method, AUGCAL increases the proportion highly-confident correct predictions (green boxes). AUG and CAL applied alone can potentially reduce that proportion (yellow boxes). AUG is PASTA, CAL is DCA.

(with PASTA as AUG and DCA as CAL) is effective in reducing SIM2REAL miscalibration across all settings. We discuss these results in Sec. A.7 of appendix.

▷ **How does AUGCAL compare with temperature scaling?** While we focus on "training-time" patches to improve SIM2REAL calibration, we also conduct an experiment to compare AUGCAL with "post-hoc" temperature scaling (TS) on VisDA. Specifically, we use 80% of VisDA SIM images for training models and rest (20%) for validation and temperature tuning. To ensure a fair comparison, we consider temperature tuning on both "clean" (C) and "PASTA augmented" (P) val splits. We find that irrespective of tuning on C or P, unlike AUGCAL, TS is ineffective and increases overconfidence and miscalibration. We present these results in Sec. A.7 of appendix.

▷ **AUGCAL increases the proportion of accurate and certain predictions.** In Fig. 3, we show qualitatively for GTAV→Cityscapes SemSeg how AUGCAL increases the proportion of highly-confident correct predictions. In practice, we find that this improvement is much more subtle for stronger SIM2REAL adaptation methods, such as HRDA + MIC, compared to weaker ones, such as EntMin + MIC, which have considerable room for improvement.

# 6 CONCLUSION

We propose AUGCAL, a method to reduce the miscalibration of SIM2REAL adapted models, often caused due to highly-confident incorrect predictions. AUGCAL modifies a SIM2REAL adaptation framework by making two minimally invasive changes – (1) augmenting SIM images via AUG transformations that reduce SIM2REAL distance and (2) optimizing for an additional calibration loss on AUGmented SIM predictions. Applying AUGCAL to existing adaptation methods for semantic segmentation and object recognition reduces miscalibration, overconfidence and improves reliability of confidence scores, all while retaining or improving performance on REAL data. AUGCAL is meant to be a task-agnostic, general purpose framework to reduce miscalibration for SIM2REAL adaptation methods and we hope such simple methods are taken into consideration for experimental settings beyond the ones considered in this paper.

**Acknowledgements.** This work has been partially sponsored by NASA University Leadership Initiative (ULI) #80NSSC20M0161, ARL and NSF #2144194.

## 7 REPRODUCIBILITY STATEMENT

We provide training, implementation and optimization details associated with our experiments in Sec. A.5 of the appendix. Most of our experiments follow the implementations of SIM2REAL adaptation methods from (Hoyer et al., 2022c), with our additional modifications on top. Additionally, in Sec. A.10 of appendix, we provide details surrounding the assets (and corresponding licenses) used for our experiments. Assumptions surrounding the analytical justification behind AUGCAL (Sec. 3.2.1 and 3.2.2) have been presented in the same sections.

## 8 ETHICS STATEMENT

Our proposed patch, AUGCAL, is meant to improve the reliability of SIM2REAL adapted models. We assess reliability in terms of confidence calibration (prediction scores aligning with true likelihood of correctness) and the extent to which calibrated confidence scores are useful for assessing prediction quality (measured via mis-classification detection). AUGCAL adapted models have promising consequences for downstream applications. A well-calibrated and reliable SIM2REAL adapted model can increase transparency in REAL predictions and facilitate robust decision making in safety-critical scenarios. That said, we would like to note that while AUGCAL is helpful for our specific measures of reliability, exploration along other domain specific notions of reliability remain.

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

# A APPENDIX

## A.1 OVERVIEW

This appendix is organized as follows. In Sec. A.4, we provide more details on the AUG choices for AUGCAL– PASTA and RandAugment. Sec. A.5 outlines training, implementation details and choice of hyperparameters. Then, in Sec. A.6, we discuss Prediction Rejection Ratio (PRR), the metric used to assess reliability in our experiments. In Sec. A.7, we provide AUGCAL results for CNN backbones, comparisons with temperature scaling and discuss sensitivity to $\lambda_{\text{CAL}}$ (coefficient of $\mathcal{L}_{\text{CAL}}$). Then, we provide more supporting qualitative examples in Sec. A.8. In Sec. A.9, we describe the SIM2REAL adaptation methods we use in detail. Finally, Sec. A.10 summarizes the assets used for our experiments and their associated licenses.

## A.2 AUG CHOICES

## A.3 PASTA

We use Proportional Amplitude Spectrum Training Augmentation (PASTA) (Chattopadhyay* et al., 2023) as one of the AUG choices in AUGCAL. PASTA applies structured perturbations (controlled by hyper-parameters $\alpha, \beta, k$) to the amplitude spectra of synthetic images to generate augmented views. For a single-channel image, $x \in \mathbb{R}^{H \times W}$ (for illustration purposes), the augmentation process in PASTA is outlined below. We refer the reader to (Chattopadhyay* et al., 2023) for more details.

1. Set $\alpha = 3.0, \beta = 0.25, k = 2$ for PASTA

2. Use FFT (Nussbaumer, 1981) to obtain the Fourier spectrum of synthetic image $x$, as $\mathcal{F}(x) = \text{FFT}(x) \in \mathbb{C}^{H \times W}$

3. Obtain the corresponding amplitude and phase spectra as $\mathcal{A}(x) = \text{Abs}(\mathcal{F}(x))$ and $\mathcal{P}(x) = \text{Ang}(\mathcal{F}(x))$ respectively.

4. Zero-center the amplitude spectrum, as $\mathcal{A}(x) = \text{FFTShift}(\mathcal{A}(x))$, to get the lowest frequency components at the center.

5. Define perturbation strength, $\sigma \in \mathbb{R}^{H \times W}$, as $\sigma[m, n] = \left(2\alpha\sqrt{\frac{m^2+n^2}{H^2+W^2}}\right)^k + \beta$, where $m, n$ denote the spatial frequencies.

6. Sample perturbations using the perturbation strength as, $\epsilon \sim \mathcal{N}(1, \sigma^2)$

7. Perturb the amplitude spectrum $\hat{\mathcal{A}}(x) = \epsilon \odot \mathcal{A}(x)$

8. Reset the low-frequency components, $\hat{\mathcal{A}}(x) = \text{FFTShift}(\hat{\mathcal{A}}(x))$

9. Obtain the augmented synthetic image via inverse FFT as, $\hat{x} = \text{iFFT}(\hat{\mathcal{A}}(x), \mathcal{P}(x))$

We set PASTA hyper-parameters as $\alpha = 3.0, \beta = 0.25, k = 2$ as it seems to work well across multiple SIM2REAL shifts in practice. We use Pytorch (Paszke et al., 2019) to vectorize all operations in the steps above and apply PASTA on minibatches, instead of individual synthetic images.

## A.4 RANDAUGMENT

RandAugment (Cubuk et al., 2020) generates augmented views by applying a series of transforms sampled from a predefined vocabulary. While the vocabulary includes geometric as well as photometric transforms, we only use photometric transforms for our experiments. For SemSeg, the models we train already use geometric transforms that are optimal for SIM2REAL SemSeg performance. For ObjRec, we find it empirically beneficial to fix geometric transforms to a `CenterCrop` and use photometric transforms from RandAugment. These choices follow (Hoyer et al., 2022c). These operations are – `AutoContrast`, `Equalize`, `Contrast`, `Brightness`, `Sharpness`, `Posterize`, `Solarize` and `SolarizeAdd`. We use ($m = 30, n = 8$) while sampling operations from this vocabulary for RandAugment. We refer the reader to (Cubuk et al., 2020) for more details.

## A.5 TRAINING AND IMPLEMENTATION DETAILS

We outline training and implementation details associated with our experiments.

**Semantic Segmentation.** For SemSeg, we use GTAV (Sankaranarayanan et al., 2018) as the source SIM dataset and Cityscapes (Cordts et al., 2016) as the target real dataset. We consider two

Table 5: **Reproducing SIM2REAL Adaptation Baselines.** We summarize results from our reproduction of the SIM2REAL adaptation baselines (from (Hoyer et al., 2022c)) used in our experiments. We find a slight difference between reported and reproduced adaptation results across models on both (a) GTAV→Cityscapes and (b) VisDA syn→real. $\pm$ indicates standard error.

| Method | Model | Status | mIoU (↑) | Method | Model | Status | mAcc (↑) |
|---|---|---|---|---|---|---|---|
| 1 HRDA + MIC | DeepLabv2 | (Hoyer et al., 2022c) | 64.20 | 1 SDAT + MIC | ResNet-101 | (Hoyer et al., 2022c) | 86.90 |
| 2 HRDA + MIC | DeepLabv2 | Reproduced | 64.05 | 2 SDAT + MIC | ResNet-101 | Reproduced | $81.03_{\pm 2.32}$ |
| 3 HRDA + MIC | DAFormer | (Hoyer et al., 2022c) | 75.90 | 3 SDAT + MIC | ViT-B | (Hoyer et al., 2022c) | 92.80 |
| 4 HRDA + MIC | DAFormer | Reproduced | 75.56 | 4 SDAT + MIC | ViT-B | Reproduced | $92.62_{\pm 0.28}$ |

(a) GTAV→Cityscapes.      (b) VisDA SIM2REAL.

segmentation architectures for our experiments – DeepLabv2 (Chen et al., 2017) (with a ResNet-101 (He et al., 2016) backbone) and DAFormer (Hoyer et al., 2022a) (with an MiT-B5 (Xie et al., 2021) backbone). Both models are initialized from backbones pretrained on ImageNet (Deng et al., 2009). For EntMin (Vu et al., 2019) (with DAFormer / DeepLabv2), we use SGD as an optimizer with a learning rate of $2.5 \times 10^{-4}$ and use $\lambda_{UDA} = 0.001$ as the coefficient of the "unconstrained" entropy loss. For HRDA, we use the multi-resolution self-training strategy from (Hoyer et al., 2022b) – AdamW (Loshchilov & Hutter, 2017) as the optimizer with a learning rate of $6 \times 10^{-5}$ for the encoder and $6 \times 10^{-4}$ for the decoder, with a linear learning rate warmup (warmup iterations 1500; warmup ratio $10^{-6}$), followed by polynomial decay (to an eventual learning rate of 0). For the teacher-student self-training setup in HRDA, we additionally couple cross-domain mixing with augmentations (DACS (Tranheden et al., 2021)) to improve self-training, use the ImageNet feature distance loss (Hoyer et al., 2022a), set $\lambda_{UDA} = 1$ and use $\alpha = 0.999$ as the teacher EMA factor. For all SemSeg settings, we use a batch size of 2 (2 source images, 2 target images) and additionally use rare class sampling (based on source; following (Hoyer et al., 2022a)) to ensure consistent adaptation improvements across all classes. For all our SemSeg experiments, we feed crops of size $1024 \times 1024$ as input to the models, irrespective of the adaptation method. PASTA for AUGCAL is applied to the same crops. We train all segmentation models for 40k iterations and use the last checkpoint to report results (the SIM2REAL adaptation setting does not assume access to labels on real data which prevents selecting "best-on-target" checkpoint). For semantic segmentation, following prior work (Wang et al., 2022), for ECE (and other associated metrics), instead of pooling all pixels of different images into a set, we compute per-image numbers and then average across images. We use the same to compute standard error. We report all metrics (performance, calibration, etc.) as percentages. We use 15 bins to compute ECE.

**Object Recognition.** For ObjRec, we conduct experiments on the VisDA (Peng et al., 2017) SIM2REAL benchmark, using standard source-target splits (Hoyer et al., 2022c) for training. We consider ImageNet (Deng et al., 2009) pretrained ResNet-101 (He et al., 2016) and ViT-B/16 (Dosovitskiy et al., 2020) backbones with standard bottleneck (with Linear, BatchNorm and ReLU) and classifier layers. As noted earlier, we use SDAT (Rangwani et al., 2022) as the adaptation method, which relies on Conditional Domain Adversarial Adaptation (CDAN) (Long et al., 2018) and Minimum Class Confusion (MCC) (Jin et al., 2020) with SAM (Foret et al., 2020) (smoothness 0.2) – all adaptation losses are combined and $\lambda_{UDA}$ is set to 1. We use SGD as the base optimizer with a learning rate of $2 \times 10^{-4}$, with a batch size of 32 (for both source and target). PASTA for AUGCAL is applied to the raw input SIM images. We train ResNet-101 backbones for 30 epochs and ViT-B/16 backbones for 15 epochs and use the last checkpoint to report results (the SIM2REAL adaptation setting does not assume access to labels on real data which prevents selecting "best-on-target" checkpoint). We report all metrics (performance, calibration, etc.) as percentages. We use 15 bins to compute ECE. For our key results in the main paper (Tables 2(b) & 3(b)), we run experiments across 3 random seeds.

**MIC Hyperparameters.** We also use MIC (Hoyer et al., 2022c) with all the discussed SIM2REAL adaptation methods since it demonstrably improves performance. For MIC, following prior work (Hoyer et al., 2022c), we use a masking patch size of 64, a masking ratio of 0.7, a loss weight of 1 and an EMA factor of 0.999 for the pseudo-label generating teacher.

**Compute.** We conduct all object recognition experiments on RTX 6000 GPUs – every experiment requiring a single GPU. For semantic segmentation, we use one A40 GPU per experiment.

Table 6: **AUGCAL ensures SIM2REAL adapted models make accurate, calibrated and reliable predictions.** We find that applying AUGCAL to multiple SIM2REAL adaptation methods across tasks leads to better calibration (ECE, IC-ECE), reduced overconfidence (OC) and improved reliability (PRR) – all while retaining or improving transfer performance. Highlighted rows are AUGCAL variants of the base methods. For AUGCAL, we use PASTA as AUG and DCA as CAL. $\pm$ indicates standard error.

| Method | Perf. ($\uparrow$) mIoU | Calibration Error ($\downarrow$) ECE | IC-ECE | OC | Reliability ($\uparrow$) PRR |
|---|---|---|---|---|---|
| 1 EntMin + MIC | 47.77 | $4.93_{\pm0.20}$ | $70.09_{\pm0.22}$ | $75.80_{\pm0.58}$ | $48.87_{\pm0.45}$ |
| 2   + AUGCAL | **49.96** | $\mathbf{3.21_{\pm0.16}}$ | $\mathbf{67.99_{\pm0.22}}$ | $\mathbf{74.33_{\pm0.57}}$ | $\mathbf{52.07_{\pm0.48}}$ |
| 3 HRDA + MIC | 64.05 | $3.52_{\pm0.19}$ | $78.94_{\pm0.18}$ | $86.48_{\pm0.50}$ | $63.20_{\pm0.48}$ |
| 4   + AUGCAL | 63.95 | $\mathbf{2.55_{\pm0.11}}$ | $\mathbf{74.71_{\pm0.21}}$ | $\mathbf{84.13_{\pm0.52}}$ | $\mathbf{65.37_{\pm0.50}}$ |

(a) GTAV→Cityscapes. (DeepLabv2 R-101).

| Method | Perf. ($\uparrow$) mIoU | Calibration Error ($\downarrow$) ECE | IC-ECE | OC | Reliability ($\uparrow$) PRR |
|---|---|---|---|---|---|
| 1 SDAT + MIC | $81.03_{\pm2.32}$ | $14.32_{\pm1.41}$ | $85.06_{\pm1.17}$ | $82.77_{\pm0.88}$ | $46.92_{\pm4.21}$ |
| 2   + AUGCAL | $\mathbf{84.27_{\pm2.22}}$ | $\mathbf{11.89_{\pm1.14}}$ | $\mathbf{82.98_{\pm0.87}}$ | $\mathbf{81.10_{\pm0.63}}$ | $\mathbf{53.26_{\pm3.52}}$ |

(b) VisDA SIM2REAL. (ResNet-101).

Table 7: **Comparing AUGCAL with Temperature Scaling for VisDA SIM2REAL.** We do a controlled experiment (with an 80-20 split of the VisDA SIM2REAL split) to compare AUGCAL with Temperature Scaling (TS). B = SDAT + MIC (ViT-B). C (clean) and P (PASTA augmented) indicate the synthetic "labeled" validation splits the temperature was tuned on. For AUGCAL, we use PASTA as AUG and DCA as CAL. $\pm$ indicates standard error.

| Method | Perf. ($\uparrow$) mIoU | Calibration Error ($\downarrow$) ECE | IC-ECE | OC | Reliability ($\uparrow$) PRR |
|---|---|---|---|---|---|
| 1 SDAT + MIC (B) | $92.24_{\pm0.30}$ | $8.13_{\pm0.36}$ | $91.54_{\pm0.05}$ | $89.48_{\pm0.43}$ | $63.92_{\pm1.46}$ |
| 1 B + TS (C) | $92.24_{\pm0.30}$ | $8.19_{\pm0.26}$ | $95.41_{\pm0.08}$ | $93.97_{\pm0.20}$ | $66.58_{\pm1.23}$ |
| 1 B + TS (P) | $92.24_{\pm0.30}$ | $8.54_{\pm0.40}$ | $93.52_{\pm0.36}$ | $92.00_{\pm0.42}$ | $64.05_{\pm1.50}$ |
| 2 B + AUGCAL | $\mathbf{92.68_{\pm0.17}}$ | $\mathbf{7.24_{\pm0.11}}$ | $\mathbf{90.40_{\pm0.09}}$ | $\mathbf{87.57_{\pm0.61}}$ | $\mathbf{66.49_{\pm0.44}}$ |

(b) AUGCAL vs Temperature Scaling.

| Method | Perf. ($\uparrow$) mIoU | Calibration Error ($\downarrow$) ECE | IC-ECE | OC | Reliability ($\uparrow$) PRR |
|---|---|---|---|---|---|
| 1 SDAT + MIC (B) | $92.24_{\pm0.30}$ | $8.13_{\pm0.36}$ | $91.54_{\pm0.05}$ | $89.48_{\pm0.43}$ | $63.92_{\pm1.46}$ |
| 2   + AUGCAL | $\mathbf{92.68_{\pm0.17}}$ | $\mathbf{7.24_{\pm0.11}}$ | $\mathbf{90.40_{\pm0.09}}$ | $\mathbf{87.57_{\pm0.61}}$ | $\mathbf{66.49_{\pm0.44}}$ |
| 1   + TS (C) | $92.68_{\pm0.17}$ | $8.01_{\pm0.13}$ | $94.44_{\pm0.17}$ | $92.78_{\pm0.53}$ | $66.10_{\pm0.65}$ |
| 1   + TS (P) | $92.68_{\pm0.17}$ | $8.02_{\pm0.12}$ | $94.46_{\pm0.20}$ | $92.80_{\pm0.58}$ | $66.05_{\pm0.65}$ |

(b) AUGCAL + Temperature Scaling.

**Reproduced Results.** We use open-sourced code for (Hoyer et al., 2022c)[3] and re-run the base adaptation methods for HRDA + MIC and SDAT + MIC at our end to obtain baseline results. In Table. 5, we summarize reported and reproduced results for the same methods.

## A.6 PREDICTION REJECTION RATIO (PRR)

As noted in Sec. 4 of the main paper, in addition to measuring reduced miscalibration, we also assess the extent to which such improvements in calibration are useful and lead to a more reliable model. To measure this, following prior work (de Jorge et al., 2023; Malinin et al., 2019), we measure whether confidence scores can reliably guide misclassification detection – measured via the PRR metric.

Ideally, in a real-world scenario, given a model, we would like to retrieve all (potentially) samples misclassified by the model based on confidence scores. These samples can then be skipped when the model is used for decision-making (since the model is likely to make incorrect predictions). Since confidence scores, in practice, are imperfect, measuring misclassification detection helps us assess this specific capability of the SIM2REAL adapted models. For a model that is less overconfident on incorrect samples (meaning it has reduced miscalibration), this specific ability should naturally be enhanced.

This can be measured using Rejection-Accuracy curves (de Jorge et al., 2023; Malinin et al., 2019) where we reject samples below a threshold and keep track of accuracy and the fraction of rejected samples. Since such curves are naturally biased towards models that have improved performance, the AUC for such a rejection curve can be normalized by that of an oracle. Additionally, we can subtract a score associated with a random baseline (sorting predictions for filtering in a random order) (Wang et al., 2021). Finally, we can compare this value (PRR; ranging from -100 to 100; higher is better) for multiple models to assess how reliable underlying confidence scores are.

## A.7 AUGCAL RESULTS

▷ **AUGCAL results with CNN architectures.** Key results presented in Table. 2 of the main paper use transformer architectures (DAFormer for SemSeg, ViT-B for ObjRec). In Tables. 6 (a) and (b), we verify that AUGCAL improvements translate to CNN based architectures as well (also discussed in Sec. 5.2 of the paper). We use DeepLabv2 (ResNet-101) for SemSeg and a ResNet-101 based classifier for ObjRec and find that applying AUGCAL improves or retains performance, reduces miscalibration and improves reliability.

▷ **Comparing AUGCAL with Temperature Scaling (TS).** While we focus on "training-time" patches to improve SIM2REAL calibration, we also conduct an experiment to compare AUGCAL with "post-hoc" temperature scaling (TS) on VisDA. Specifically, we use 80% of VisDA SIM images for training

---

[3]https://github.com/lhoyer/MIC

Table 8: **Sensitivity to $\lambda_{\text{CAL}}$ for AUGCAL on GTAV→Cityscapes.** We vary the value of $\lambda_{\text{CAL}} \in \{0.1, 0.5, 1.0, 5.0, 10.0, 20.0, 100.0\}$ and report the effect on adaptation performance, reduced miscalibration and improved reliability. We find that our choice of $\lambda_{\text{CAL}} = 1$ leads to balanced performance across desired metrics. Rows in red correspond to the baseline SIM2REAL adaptation method without the application of AUGCAL.

| Value of $\lambda_{\text{CAL}}$ | Perf. (↑) | Calibration Error (↓) | | Reliability (↑) |
| | mIoU | ECE | IC-ECE | PRR |
| --- | --- | --- | --- | --- |
| 1 No AUGCAL | 65.71 | 5.34 | 77.73 | 45.93 |
| 2 $\lambda_{\text{CAL}} = 0.1$ | 69.72 | 2.88 | 74.55 | 57.06 |
| 3 $\lambda_{\text{CAL}} = 0.5$ | 69.27 | 2.71 | 73.54 | 60.32 |
| 4 $\lambda_{\text{CAL}} = 1.0$ | 70.31 | 3.43 | 72.97 | 62.66 |
| 5 $\lambda_{\text{CAL}} = 5.0$ | 66.24 | 3.55 | 70.22 | 62.66 |
| 6 $\lambda_{\text{CAL}} = 10.0$ | 64.27 | 3.28 | 67.63 | 64.55 |
| 7 $\lambda_{\text{CAL}} = 20.0$ | 55.37 | 4.17 | 65.55 | 63.20 |
| 8 $\lambda_{\text{CAL}} = 100.0$ | 27.73 | 6.06 | 52.24 | 54.47 |

(a) EntMin + MIC (DAFormer).

| Value of $\lambda_{\text{CAL}}$ | Perf. (↑) | Calibration Error (↓) | | Reliability (↑) |
| | mIoU | ECE | IC-ECE | PRR |
| --- | --- | --- | --- | --- |
| 1 No AUGCAL | 75.56 | 2.86 | 81.92 | 68.91 |
| 2 $\lambda_{\text{CAL}} = 0.1$ | 75.25 | 2.94 | 80.68 | 69.96 |
| 3 $\lambda_{\text{CAL}} = 0.5$ | 75.05 | 2.77 | 80.23 | 70.62 |
| 4 $\lambda_{\text{CAL}} = 1.0$ | 75.90 | 2.45 | 79.09 | 70.35 |
| 5 $\lambda_{\text{CAL}} = 5.0$ | 73.80 | 2.25 | 76.61 | 70.54 |
| 6 $\lambda_{\text{CAL}} = 10.0$ | 71.28 | 2.50 | 75.85 | 69.17 |
| 7 $\lambda_{\text{CAL}} = 20.0$ | 62.31 | 2.43 | 73.32 | 68.27 |
| 8 $\lambda_{\text{CAL}} = 100.0$ | 31.01 | 7.68 | 73.56 | 56.52 |

(a) HRDA + MIC (DAFormer).

models and rest (20%) for validation and temperature tuning. To ensure a fair comparison, we consider temperature tuning on both "clean" (C) and "PASTA augmented" (P) val splits.[4] We present these results in Table. 7 (a). We find that irrespective of tuning on C or P, unlike AUGCAL, TS is ineffective and increases overconfidence and miscalibration. In Table. 7, we additionally consider temperature scaling on the logits of an AUGCAL improved SIM2REAL model and find that it worsens miscalibration and overconfidence. Note that this is not entirely surprising since TS depends heavily on the "data-split" the temperature is tuned on, which in our case is SIM and not REAL.

▷ **Sensitivity to $\lambda_{\text{CAL}}$.** For an existing SIM2REAL adaptation pipeline, AUGCAL involves optimizing for an additional calibration loss $\mathcal{L}_{\text{CAL}}$ on augmented synthetic images. In Table. 8, we vary the coefficient $\lambda_{\text{CAL}}$ (values in the set $\{0.1, 0.5, 1.0, 5.0, 10.0, 20.0, 100.0\}$) for $\mathcal{L}_{\text{CAL}}$ and note the effect on adaptation performance, calibration on real data (ECE, IC-ECE) and reliability (PRR). In Table 2 of the main paper, we already note how applying AUGCAL with $\lambda_{\text{CAL}} = 1$ improves over a baseline adaptation method (red and blue rows in Table. 8). We further note that compared to other values, our choice of $\lambda_{\text{CAL}} = 1.0$ achieves a balance between adaptation performance, reduced miscalibration and improved reliability. We find that overly high values of $\lambda_{\text{CAL}} \geq 5$ can potentially lead to reduced adaptation performance – $\lambda_{\text{CAL}} \geq 5$ significantly raises the scale of $\mathcal{L}_{\text{CAL}}$ compared to $\mathcal{L}_{CE}$ and $\mathcal{L}_{UDA}$, which leads to models optimizing for improved calibration at the expense of task performance. Based on our experiments across multiple models, shifts and tasks, we recommend restricting $\lambda_{\text{CAL}} < 5$ for SIM2REAL adaptation.

## A.8    QUALITATIVE PREDICTIONS

In Figures 4 and 5, we provide more examples to demonstrate how AUGCAL improves the proportion of "accurate" and "certain" predictions. In Fig. 4, where we compare predictions for a (DAFormer) HRDA + MIC model – w/o AUGCAL 75.56 mIoU and w AUGCAL 75.90 mIoU. We find that applying AUGCAL improves the proportion of "accurate" and "certain" predictions (confidence > 0.95) – see Fig. 4, columns 3 and 5, guided by yellow arrows. For instance, we find that the base model (w/o AUGCAL) has trouble assigning high-confidence to correct "sidewalk" and "vegetation" predictions. For EntMin + MIC (w/o AUGCAL 65.71 mIoU and w AUGCAL 70.31 mIoU) in Fig. 5, we find that AUGCAL is considerably more effective in ensuring high-confidence correct predictions. Notably, we find these improvements to be more subtle as the SIM2REAL adaptation method itself improves (HRDA + MIC > EntMin + MIC).

## A.9    SIM2REAL ADAPTATION METHODS

We conduct experiments with three SIM2REAL adaptation methods across two tasks. We wanted to assess the compatibility of AUGCAL with SIM2REAL adaptation methods that are competitive and are representative of the broader class of approaches used for SIM2REAL transfer. We first cover some background and describe these methods in detail.

---

[4]Note that temperature tuning (by definition) only affects calibration and has no impact on performance.

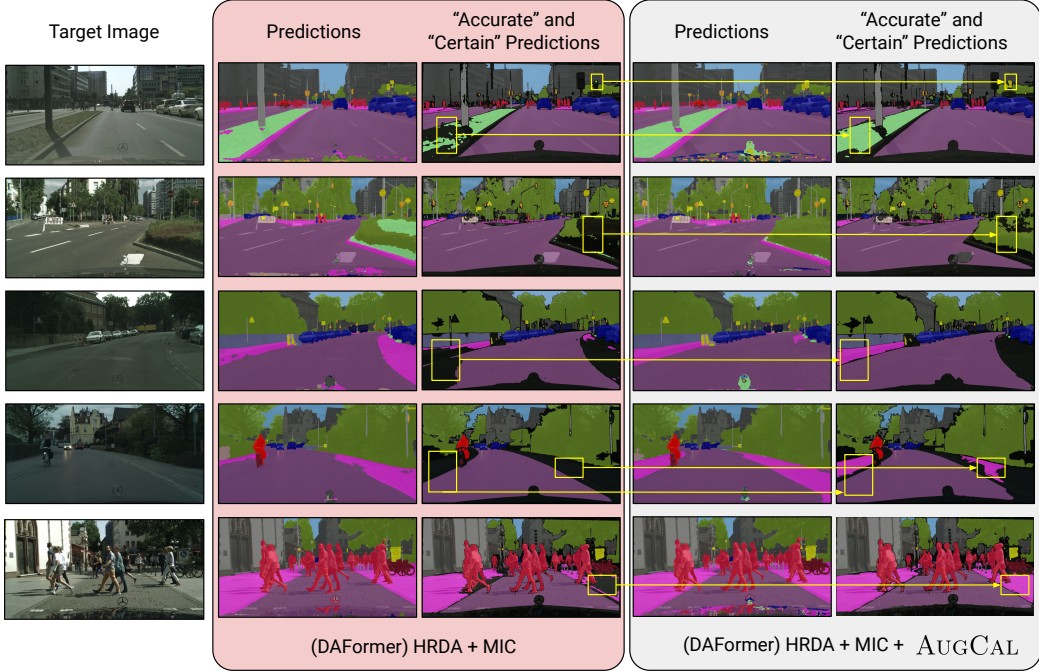

Figure 4: **AUGCAL increases the proportion of "accurate" and "certain" predictions**. For a base DAFormer SemSeg model trained with HRDA + MIC (State-of-the-art) on GTAV→Cityscapes, we show how applying AUGCAL at training time (right) can improve the proportion of "accurate" and "certain" (confidence > 0.95) predictions over the vanilla adaptation method (left). Regions in black do not satisfy the "accurate" and "certain" filtering criteria.

In unsupervised domain adaptation (UDA) for SIM2REAL settings, we assume access to a labeled (SIM) source dataset $D_S = \{(x_i^S, y_i^S)\}_{i=1}^{|S|}$ and an unlabeled (REAL) target dataset $D_T = \{x_i^T\}_{i=1}^{|T|}$. We assume $D_S$ and $D_T$ splits are drawn from source and target distributions $P^S(x, y)$ and $P^T(x, y)$ respectively. At training, we have access to $D = D_S \cup D_T$. We operate in the setting where source and target share the same label space, and discrepancies exist only in input images. The model $\mathcal{M}_\theta$ is trained on labeled source images using cross entropy,

$$\sum_{i=1}^{|S|} \mathcal{L}_{CE}(x_i^S, y_i^S; \theta) = -\sum_{i=1}^{|S|} y_i^S \log p_\theta(\hat{y}_i^S | x_i^S) \text{ where } \hat{y}_i^S = \arg\max_{y \in \mathcal{Y}} p_\theta(y_i^S | x_i^S) \quad (14)$$

For object recognition, we represent the labels corresponding to images $x \in \mathbb{R}^{H \times W \times 3}$ as $y \in \mathcal{Y}$ where $\mathcal{Y} = \{1, 2, ..., K\}$. For semantic segmentation, since we make predictions across a vocabulary of classes for every pixel, the corresponding label can be expressed as $y \in \mathcal{Y}^{H \times W}$.

UDA methods additionally optimize for an adaptation objective on labeled source and unlabeled target data ($\mathcal{L}_{UDA}$). The overall learning objective can be expressed as,

$$\min_\theta \underbrace{\sum_{i=1}^{|S|} \mathcal{L}_{CE}(x_i^S, y_i^S; \theta)}_{\text{Source Loss}} + \underbrace{\sum_{i=1}^{|T|} \sum_{j=1}^{|S|} \lambda_{UDA} \mathcal{L}_{UDA}(x_i^T, x_j^S, y_j^S; \theta)}_{\text{Source Target Adaptation Loss}} \quad (15)$$

Different adaptation methods usually differ in terms of specific instantiations of this objective. As adaptation methods, we use HRDA (Hoyer et al., 2022b) and EntMin (Vu et al., 2019) for segmentation and SDAT (Rangwani et al., 2022) for recognition.

**Entropy Minimization (EntMin).** We consider the unconstrained direct entropy minimization approach from (Vu et al., 2019) as one of the adaptation methods. EntMin builds on top of the assumption the models trained only on source data tend to be under-confident (make high-entropy predictions) on target images. EntMin enforces high prediction certainty on target images by ensuring

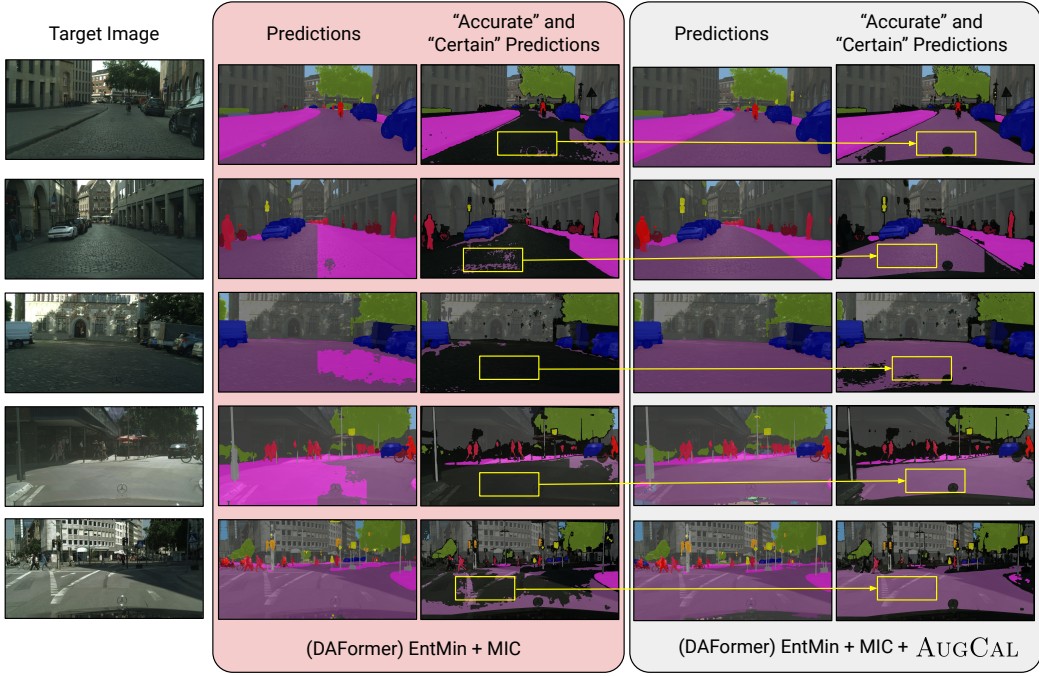

Figure 5: **AUGCAL increases the proportion of "accurate" and "certain" predictions**. For a base DAFormer SemSeg model trained with EntMin + MIC on GTAV→Cityscapes, we show how applying AUGCAL at training time (right) can improve the proportion of "accurate" and "certain" (confidence $> 0.95$) predictions over the vanilla adaptation method (left). Regions in black do not satisfy the "accurate" and "certain" filtering criteria.

that the model makes high-confidence predictions on the same. This is realized by minimizing the normalized entropy of target predictions. Specifically, given a target image $x^T$, if the predictive probabilities per-pixel are expressed as $\mathbf{P}_{x^T}^{(h,w,k)} \in [0,1]^{H \times W \times K}$, the adaptation loss on target can be expressed as,

$$\mathcal{L}_{UDA}(x^T; \theta) = \frac{1}{HW} \sum_{h,w} \frac{-1}{\log K} \sum_{k=1}^{K} \mathbf{P}_{x^T}^{(h,w,k)} \log \mathbf{P}_{x^T}^{(h,w,k)} \qquad (16)$$

EntMin can also be viewed as a "soft-assignment" version of self-training (Lee et al., 2013; Zou et al., 2018). We refer the reader to (Vu et al., 2019) for more details.

**Context-Aware High-Resolution Domain-Adaptive Semantic Segmentation (HRDA).** The base adapation method in HRDA (Hoyer et al., 2022b) is self-training (Zou et al., 2018; Lee et al., 2013). HRDA additionally introduces components that are beneficial for domain-adaptive semantic segmentation. Specifically, HRDA proposes a multi-resolution framework by relying on (1) a low-resolution *context* crop to learn long-range contextual dependencies and (2) a high-resolution *detail* crop to make detailed and accurate predictions. During adaptation, HRDA fuses predictions from both crops using input dependent attention. For a target image $x^T$, pseudo-labels are obtained from teacher network $\mathcal{M}_\phi$ (a moving average of $\mathcal{M}_\theta$) as $\mathbf{P}_{x^T}^{(h,w)} = \arg\max_k \mathcal{M}_\phi(x^T)$. HRDA adapts to target by minimizing cross entropy w.r.t. $\mathbf{P}_{x^T}^{(h,w)}$ as,

$$\mathcal{L}_{UDA}(x^T; \theta) = -\sum_{h,w} q_{x^T} \mathbf{P}_{x^T}^{(h,w)} \log \mathcal{M}_\theta(x^T) \text{ with } q_{x^T} \text{ being a quality estimate for } \mathbf{P}_{x^T}^{(h,w)} \quad (17)$$

Instead of "self"-training on target images, HRDA uses cross-domain mixing (DACS (Tranheden et al., 2021)) to obtain pseudo-labels on augmented images. Additionally, components from DAFormer (Hoyer et al., 2022a) – rare class sampling and an ImageNet feature distance loss – are incorporated in the pipeline to facilitate better adaptation. The quality estimate $q_{x^T}$ is computed as the proportion of pixels that have confidence above a specified threshold. This naturally ensures a warmup stage, where a model is first trained only on synthetic images for a few iterations, followed

by training on both synthetic and real images. We refer the reader to (Hoyer et al., 2022b) for more low-level implementation details.

**Smooth Domain Adversarial Training (SDAT).** SDAT (Rangwani et al., 2022) is an adaptation method for object recognition. The underlying adaptation method in SDAT is domain adversarial training (DAT), which involves reducing the discrepancy between source and target image distributions. This is realized by confusing an additional discriminator that is designed to distinguish between source and target samples. While multiple versions of DAT exist, SDAT uses CDAN (Long et al., 2018) as the default DAT method. SDAT investigates the loss-landscapes of DAT style methods and notes that smoother loss-landscapes on source data result in improved transfer to target. Consequently, SDAT proposes optimizing for smoother loss landscapes on labeled source data by modifying the supervised $\mathcal{L}_{CE}$ loss as,

$$\sum_{i=1}^{|S|} \mathcal{L}_{CE}(x_i^S, y_i^S; \theta) = \sum_{i=1}^{|S|} y_i^S \max_{||\epsilon|| \leq \rho} \log p_{\theta+\epsilon}(\hat{y}_i^S | x_i^S) \quad (18)$$

Perturbing the weights of the network $\theta$ by $\epsilon$ in some neighborhood $\rho$ ensures lower loss values in that neighborhood, thereby encouraging a smoother loss landscape. In practice, this is realized by using sharpness aware minimization (SAM) (Foret et al., 2020) to update model parameters. We refer the reader to (Rangwani et al., 2022; Long et al., 2018) for more details on the UDA losses used.

**Masking Image Consistency (MIC).** MIC (Hoyer et al., 2022c) is a general technique applicable to any existing adaptation method to improve context utilization while making predictions. MIC involves a teacher-student self-training setup where pseudo-labels are obtained from a weakly augmented target sample (via the teacher). Student predictions for a masked image are forced to match the obtained pseudo-labels. MIC relies on the recent success of masking as an auxiliary task (He et al., 2022) but instead of reconstruction, MIC sets up a prediction consistency task. This simple addition to an existing adaptation pipeline leads to considerable improvements across multiple tasks and shifts. A "masked" image for MIC is obtained by dividing the original image into patches and randomly masking a subset of the same. Similar to HRDA (Hoyer et al., 2022b), the MIC (consistency) loss is multiplied by a quality estimate of the pseudo-labels. We refer the reader to (Hoyer et al., 2022c) for more details on MIC.

## A.10 ASSETS AND LICENCES

The assets used in this work can be grouped into three categories – Datasets, Code Repositories and Dependencies. We discuss source and licences for each of these below.

**Datasets.** For semantic segmentation, we use the GTAV (Richter et al., 2016) and Cityscapes (Cordts et al., 2016) datasets. Code used to extract densely annotated images from the GTAV game is distributed under the MIT license.[5] The Cityscapes' license agreement dictates that the dataset is made freely available to academic and non-academic entities for non-commercial purposes such as academic research, teaching, scientific publications, or personal experimentation and that permission to use the data is granted under certain conditions.[6] For object recognition, we use the VisDA syn→real (Peng et al., 2017) benchmark. The VisDA-C development kit on github does not have a license associated with it, but it does include a Terms of Use, which primarily states that the dataset must be used for non-commercial and educational purposes only.[7]

**Code Repositories.** For our experiments, apart from code that we wrote ourselves, we build on top of the open-sourced codebase for MIC (Hoyer et al., 2022c)[8] repository. MIC is distributed under the MIT License.

**Dependencies.** We use Pytorch (Paszke et al., 2019) as the deep-learning framework for all our experiments. Pytorch, released by Facebook, is distributed under a Facebook-specific license.[9]

---

[5] https://bitbucket.org/visinf/projects-2016-playing-for-data/src/master/
[6] https://www.cityscapes-dataset.com/license/
[7] https://github.com/VisionLearningGroup/taskcv-2017-public/tree/master/classification
[8] https://github.com/lhoyer/MIC
[9] https://github.com/pytorch/pytorch/blob/master/LICENSE

