# OpenReview forum: "AUGCAL: Improving Sim2Real Adaptation by Uncertainty Calibration on Augmented Synthetic Images"
_ICLR.cc/2024/Conference — ICLR 2024 poster_

### Official Review · Reviewer_WuQd · 2023-10-31

**Soundness:** 3 good
**Presentation:** 3 good
**Contribution:** 4 excellent
**Rating:** 8
**Confidence:** 4

**Summary:**

This paper proposes a general Sim2Real adaptation framework, AUGCAL, for semantic segmentation and object recognition.
AUGCAL introduces synthetic data augmentation and model calibration on synthetic data to reduce miscalibration during synthetic training. Combining with standard unsupervised domain adaptation methods on unlabeled real-world data, those techniques further improve SIM2REAL performance on real-world data.
Especially, this paper provides theoretical analysis to show the target calibration loss can be bound by the source calibration loss. This is the reason that calibration on synthetic data is beneficial for the performance of real-world data. Also, it shows the necessity of introducing the augmentaiton.
Experiments are conducted on GTAV to Cityscapes and VisDA SIM2REAL, and show that the proposed framework can improve the baselines easily.

**Strengths:**

[Clarity] The paper is well structured, and I find myself easy to follow.

[Baseline] Baselines with AUGCAL outperform those without AUGCAL in all metrics.

[Implementation] The presented method looks straightforward yet effective and easily re-implementable.

[Novelty]
- Augmentation and calibration are complementary and can be integrated easily to achieve the goal.
- Miscalibration is an overlooked factor for unsupervised domain adaptation, and this paper proposes an easy way to handle it.
- The paper provides insights on how the calibration loss on synthetic data can reduce miscalibration on real data via a theoretical analysis. This is beneficial for other related research fields.

**Weaknesses:**

I listed some suggestions in Questions.

**Questions:**

[Augmentation] The choice of augmentation should satisfy the property 1 in Sec. 3.2.2. Even Tab.1 shows that PASTA/R.Aug-SIM are closer to Real than SIM based on MMD, it is not intuitive. It would be better to clarify it more.
- For PASTA, it is designed to bridge the syn-to-real gap and maybe reduce the artifacts of synthetic data. But I am not sure why R.Aug even closer to Real.
- Also, if we have large-scale unlabeled real-world data, I am wondering if style transfer-based augmentation will have better performance because it directly uses unlabeled real-world data.
- Last, as property 1 is a distribution distance, I am not sure how those appearance augmentations affected distribution.

[OC] After Eq.3 and in Sec.4, the paper introduces overconfidence (OC) as a metric. But I am not sure of the exact definition.

[Eq.9] Eq.9 shows a useful upper bound. It would be more interesting to provide more insights regarding this bound. For example, what issues (e.g., a large gap between syn and real) and why will those issues lead to the loose of the upper bound?

---

> ### Author Response · Authors · 2023-11-17
> **Thanks for the feedback!**
>
> Thanks for providing feedback on our submission. We address individual concerns below.
>
> **PASTA vs RandAug distributional distances**
>
> PASTA was designed specifically to tackle Sim2Real discrepancies by introducing increased perturbations to high-frequency synthetic image components. RandAug, on the other hand, has no such constraints, and can generally perturb all (low, high frequency) components indirectly using chained photometric operations. From the measured distributional distances in Table 1 of main paper, PASTA and RandAug are similarly effective in reducing distributional distance on GTAV$\to$City, whereas RandAug is more effective than PASTA on VisDA-C. We believe this is due to the nature of VisDA-C synthetic images. Synthetic images in VisDA-C are snapshots of CAD models of object classes from different viewpoints, lighting conditions in clear backgrounds. Consequently, increased focus towards high-frequency components (PASTA) in such settings could be less beneficial compared to peturbing all frequency components (RandAug). Using both AUG choices (Sim2Real specific and otherwise) allowed us to better demonstrate the general applicability of AUGCAL for Sim2Real settings.
>
> **Would style-transfer operations be eligible AUG choices?**
>
> Good point! While a style-transfer operation that can translate synthetic images to real style (while preserving semantics) can help reduce distributional distance, it cannot perfectly mitigate the distributonal gap. This is because Sim2Real discrepancies exist not only in terms of appearance, but also content. For instance, compared to Cityscapes, synthetic scenes from GTAV tend to be much less populated with vehicle and human instances. Similarly, synthetic source images in VisDA-C contain clear backgrounds (no surrounding context) unlike the real target counterparts.
>
> Further, combining deep network (DNN) based style-transfer operations with state-of-the-art Sim2Real UDA methods is non-trivial since -- (1) it necessitates pre-training a translation network for every Sim2Real shift, (2) running DNN based style transfer online in conjunction with UDA is expensive and (3) doing such translations for synthetic source images offline would require quality checks for different Sim2Real shifts. Our intent behind experimenting with PASTA and RandAug as Aug transforms in AUGCAL was to consider simple eligible transforms (based on properties 1 and 2) that are inexpensive to combine with Sim2Real UDA methods while having shown demonstrable benefits in such settings (as shown in [A]).
>
> [A] - PASTA: Proportional amplitude spectrum training augmentation for syn-to-real domain generalization, ICCV 2023.
>
> **How do appearance augmentations affect marginal distributions?**
>
> Thanks for bringing this up. Property 1 is based on distributional distances of the marginal input (image) distributions. Assuming covariate shift conditions, we parameterize this (and measure in Table 1) in terms of image feature distances. Since appearance augmentations on images impact features, they in turn affect feature distances or equivalently distributional distances.
>
> **Insights on target calibration error bound**
>
> Thanks for the suggestion! The tightness of the bound presented in Eq. (9) depends on two key conditions -- (1) the extent to which the model is calibrated on labeled source data, (2) the extent to which the synthetic data distribution differs from the real data distribution. The first condition is straightforward to interpret -- calibration on labeled source synthetic data is necessary for calibration on real target data. The second condition, governed by the distributional distance between source and target, dictates the extent to which the stated bound is practically useful. Interventions like AUGCAL are useful when source and target marginal distributions differ substantially. If source synthetic data is highly similar to real data (in terms of appearance and content), then such interventions are unlikely to improve calibration on real data substantially.

---

> > ### Comment · Reviewer_WuQd · 2023-11-22
> >
> > Thanks for the feedback. My concerns have been addressed.

---

### Official Review · Reviewer_dFNn · 2023-10-31

**Soundness:** 3 good
**Presentation:** 3 good
**Contribution:** 3 good
**Rating:** 6
**Confidence:** 4

**Summary:**

This paper starts with a nice theory demonstrating that to achieve better calibration loss in target domain, one should minimize the miscalibration in source domain and reduce the distributional distance between source and target domains. Then, to address this, the paper proposes AUGCAL, which augments the source data and apply a calibration loss on it. Experiments demonstrate the effectiveness of the proposed method.

**Strengths:**

1. The theory is nice and understandable.
2. The paper is well-written and experiments are extensive.

**Weaknesses:**

1. Usually adding calibration loss will have lower ECE but also lower accuracy (in your case mIoU). Can you give some intuition in why adding calibration loss on source give better mIoU as shown in Table 3 (a)?
2. This paper proposes two properties and empirically verifies PASTA and RandAugment satisfy the criterion. It would be good to give some intuition in the main text.
3. [1] uses StyleNet which learns a style transfer from source to target, it would be interesting to analyze whether this inherently satisfy the property 1.
[1] Donghyun Kim, Kaihong Wang, Kate Saenko, Margrit Betke, and Stan Sclaroff. A unified framework for domain adaptive pose estimation. In ECCV. Springer, 2022.
4. Do the two AUG choice meet the property 2? Is the \epsilon as small as before AUG?
55. Since equation 9 is the upper bound of calibration loss, how is this related to mIoU?

**Questions:**

please see weaknesses

---

> ### Author Response · Authors · 2023-11-17
> **Thanks for the feedback!**
>
> Thanks for providing feedback on our submission. We address individual concerns below.
>
> **Isolated CAL Improvements**
>
> Good catch! For in-distribution settings, an additional training time calibration loss ($\mathcal{L}\_{\text{CAL}}$) doesn't necessarily compromise accuracy for reduced ECE (as shown for DCA, MDCA, MbLS). In our Sim2Real UDA setups, especially for methods that rely on self-training (or entropy minimization), errors on real data often stem from overconfident predictions. We believe adding only a training time calibration loss ($\mathcal{L}\_{\text{CAL}}$) can help reduce overconfidence, leading to better pseudo-labels during adaptation, which can *potentially* improve adaptation performance in some cases. However, this behavior of minor improvements with only CAL is not universally applicable across settings, unlike AUGCAL. Do let us know if this helps answer your question.
>
> **Justifying AUG choices**
>
> Thanks for bringing this up. Our intent was to choose AUG transformations that (1) satisfy the properties in Sec 3.2.2, and additionally, (2) are inexpensive when combined with Sim2Real UDA methods, and (3) have demonstrable benefits for Sim2Real shifts. We selected PASTA as it was **specifically** designed for Sim2Real shifts (by tackling high-frequency disparities) and RandAug because it has proven to be **generally** beneficial for Sim2Real shifts (through a series of photometric transforms, also shown in [A]). Using both AUG choices allowed us to better demonstrate the general applicability of AUGCAL. We have updated the paper to include this (in Sec. 3.2.2 in red).
>
> [A] - PASTA: Proportional amplitude spectrum training augmentation for syn-to-real domain generalization, ICCV 2023.
>
> **Do AUG choices satisfy property 2?**
>
> Yes! Both AUG choices satisfy property 2. In the Table below, using DCA as $\mathcal{L}\_{\text{CAL}}$, we log the value of $\mathcal{L}\_{\text{CAL}}$ on source data at UDA training convergence for segmentation (GTA$\to$Cityscapes) and classification (VisDA-C). We compute $\mathcal{L}\_{\text{CAL}}$ at training convergence by tracking a moving average over a window of the past 500 iterations (equivalent to 16k source training samples for VisDA-Syn and 1k 1024 $\times$ 1024 training crops for GTAV). Source $\mathcal{L}\_{\text{CAL}}$ for AUGCAL (both AUG choices) is only marginally higher than CAL. DCA measures the absolute mean difference between confidence and accuracy, showing that for these cases on source data, aggregate prediction accuracies and confidence scores differ by approximately $\sim0.01-0.02$ absolute points.
>
> |  | Shift | Sim2Real UDA Method | Setting | $\mathcal{L}\_{\text{CAL}}$ at Convergence |
> | -------- | -------- | -------- | -------- | -------- |
> |   1   | G$\rightarrow$C  | HRDA + MIC (DAFormer)    | CAL (DCA) | 0.01387 |
> |   2   | G$\rightarrow$C  | HRDA + MIC (DAFormer)    | AUGCAL (PASTA, DCA)  | 0.01473 |
> |   3   | G$\rightarrow$C  | HRDA + MIC (DAFormer)    | AUGCAL (RandAug, DCA)  | 0.01979 |
> |   4   | VisDA-C  | SDAT + MIC (ViT-B)     | CAL (DCA) | 0.00800 |
> |   5   | VisDA-C  | SDAT + MIC (ViT-B)    | AUGCAL (PASTA, DCA) | 0.01400 |
> |   6   | VisDA-C  | SDAT + MIC (ViT-B)    | AUGCAL (RandAug, DCA) | 0.01800 |
>
> **How is Eq.(9) related for performance (mIoU)?**
>
> AUGCAL aims to improve Sim2Real calibration while preserving transfer (UDA) performance. By optimizing for a calibration loss ($\mathcal{L}\_{\text{CAL}}$) alongside supervised source ($\mathcal{L}\_{\text{CE}}$) and unsupervised adaptation ($\mathcal{L}\_{\text{UDA}}$) losses, it prevents excessive optimization for $\mathcal{L}\_{\text{CE}}$ (or negative log-likelihood), a key contributor to miscalibration and overconfidence in model predictions. This approach ensures better-calibrated confidence scores on real target data (as indicated by equation 9), influencing prediction correctness (accuracy and mIoU). However, maintaining a balance between $\mathcal{L}\_{\text{CE}}$ and $\mathcal{L}\_{\text{CAL}}$ is crucial, as overly emphasizing $\mathcal{L}\_{\text{CAL}}$ during UDA + AUGCAL training (extreme $\lambda\_{\text{CAL}}\geq 5$ values) can negatively impact transfer performance, as outlined in Table 4 of the appendix. Do let us know if this helps answer your question.
>
> **Would learned style-transfer satisfy property-1?**
>
> Thanks for the pointer. Yes. A style-transfer operation (using StyleNet) capable of translating synthetic images to real style while preserving semantics can satisfy property 1. However, in practice, such a transformation cannot completely eliminate the distributional gap since Sim2Real differences exist not only in terms of appearance, but also content. For instance, compared to Cityscapes, synthetic scenes from GTAV tend to be much less populated with vehicle and human instances. We further highlight in the general response why simple yet effective AUG choices are better suited compared to learned style transfer operations for Sim2Real UDA + AUGCAL training.

---

> > ### Comment · Reviewer_dFNn · 2023-11-28
> > **Discussion**
> >
> > Thank you for your effort in rebuttal! My concerns are well addressed and I would keep my rating as 6: marginally above the acceptance threshold.

---

### Official Review · Reviewer_iaFo · 2023-11-20

**Soundness:** 3 good
**Presentation:** 3 good
**Contribution:** 2 fair
**Rating:** 6
**Confidence:** 3

**Summary:**

This paper proposes AUGCAL to improve the confidence calibration and model reliability of existing unsupervised domain adaptation approaches. Specifically, the authors propose to replace the original images with strongly augmented ones and additionally optimize for calibration loss on the augmented images. Detailed analytical justification has been derived to show how AUGCAL can reduce miscalibration on real data. Extensive experiments on different datasets and different UDA backbones have been conducted to show the effectiveness of AUGCAL.

**Strengths:**

1. Extensive experiments on different tasks, datasets and UDA backbones are provided in the paper to evaluate the proposed method.

2. Theoretical derivations have been provided to motivate the design of AUGCAL.

3. The proposed AUGCAL is simple, which only introduces two small changes to the existing UDA pipelines.

**Weaknesses:**

1. The technical contribution of this paper is a bit limited. The key components of AUGCAL are the data augmentation and DCA based calibration loss. However the data augmentation technique is commonly used in domain adaptation and the DCA proposed by previous work for model calibration.

2. The proposed method is only compared with diffident backbone UDA models. A more convincing comparison would be to compare with existing confidence calibration methods, such as [Wang et al. 2022], [Gong et al. 2021].

3. The paper exceeds the page limit of 9 pages.

**Questions:**

Please refer to the weaknesses.

---

> ### Author Response · Authors · 2023-11-20
> **Thanks for the feedback!**
>
> Thanks for the comments. We address individual concerns below.
>
> **Limited Technical Contributions**
>
> We would like to clarify our key contributions. The intent behind our proposed method, AUGCAL, was to reduce miscalibration of Sim2Real adaptation methods while preserving transfer performance. We first make the observation that errors by Sim2Real adaptation methods on "real" data are tied with increasing overconfidence on mispredictions. Based on the established goal and our observations,
>
> - We first provide a simple analytical justification surrounding how training time calibration interventions (CAL) on labeled "synthetic" source data can be helpful in reducing miscalibration on "real" target data. Building on top of this, we further demonstrate how incorporating **eligible** AUG transformations in this vanilla CAL setup (to form AUGCAL), can be even more effective in reducing Sim2Real miscalibration.
> - We empirically validate the effectiveness of AUGCAL by showing that it reduces Sim2Real miscalibration while retaining performance across multiple tasks, adaptation methods, intra-task backbones (CNNs & Transformers), and shifts. Additionally, we also assess AUGCAL effectiveness across multiple AUG (PASTA, RandAug) and CAL (DCA, MDCA, MbLS) choices.
>
> We would like to re-emphasize that the key technical contribution in AUGCAL lies precisely in demonstrating that a combination of eligible (in accordance with the properties in Sec. 3.2.2) AUG transforms with training time CALibration losses is effective in reducing Sim2Real miscalibration across multiple settings while being easy to integrate (in a plug-and-play fashion) in existing setups.
>
> **Limited Comparisons**
>
> Thanks for the suggestion. In out-of-distribution settings, both [A,B] study "*post-hoc calibration methods*" (based on temperature-scaling) for "*zero-shot generalization*" (when target data is **unavailable**). In contrast, we study "*training time-calibration methods*" (calibration losses at training time) for "*unsupervised adaptation*" (when modestly sized unlabeled target data is **available**). We note that the distinction between the two experimental settings is crucial, since unlike zero-shot settings (key experiments in [A], Sec 6.1.2 results in [B]), adaptation scenarios consist of repeated model updates based on intermediate miscalibrated target predictions. This makes adaptation scenarios much more sensitive to progressive miscalibration on target data.
>
> That said, in addition to demonstrating the effectiveness of AUGCAL across different Sim2Real adaptation models (across tasks, UDA methods, intra-task backbones, and shifts), we also conduct experiments to assess AUGCAL effectiveness for different CAL choices ([C, D, E]; which are effective confidence calibration techniques on their own). Given the limited timeframe (review received two days before end of discussion period), we are unable to perform a new equivalent comparison (by adapating [A,B] to our experimental setting).  We would like to instead point the reviewer to our temperature-scaling (TS) comparisons on VisDA-C (in Sec 5.2 main paper; Sec. 5 appendix) where we show that TS methods overfit substantially to improved calibration on "synthetic" data, making them ineffective for improving calibration on "real" data for Sim2Real adaptation methods.
>
> [A] - Confidence Calibration for Domain Generalization under Covariate Shift (Gong et. al., 2021)
>
> [B] - On Calibrating Semantic Segmentation Models: Analyses and An Algorithm (Wang et. al., 2022)
>
> [C] - Improved trainable calibration method for neural networks on medical imaging classification, BMVC 2020
>
> [D] - A stitch in time saves nine: A train-time regularizing loss for improved neural network calibration, CVPR 2022
>
> [E] - The devil is in the margin: Margin-based label smoothing for network calibration, CVPR 2022
>
> **Manuscript Length**
>
> The current revised submission (based on suggestions from R-dFNn, R-WuQd) is contained with 9 pages for the main text, with the reproducibility and ethics statement being on page 10. The author guide (see [F]) clearly specifies that both the ethics and reproducibility statements do not count towards the page limit.
>
> [F] - https://iclr.cc/Conferences/2024/AuthorGuide

---

> > ### Comment · Reviewer_iaFo · 2023-11-22
> >
> > Thanks for the authors' responses, which resolves most of my concerns. I will further discuss with other reviewers and change my rating accordingly.

---

### Author Response · Authors · 2023-11-17
**Thanks for the thoughtful comments and suggestions!**

We thank the reviewers for their thoughtful comments and constructive suggestions! We are pleased that they found the problem setting important (R-WuQd), AUGCAL to be novel, effective and easy to implement (R-WuQd), surrounding analytical justification understandable (R-dFNn), insightful and beneficial for related research (R-WuQd), our paper well-written (R-dFNn), well-structured (R-WuQd) and our experiments extensive (R-dFNn).

A recurring question from the reviewers was surrounding the applicability of style-transfer operations as AUG choices (that satisfy properties 1 and 2 in Sec. 3.2). We generally agree that a style-transfer operation that can translate synthetic images to real style (while preserving semantics) can indeed satisfy properties 1 and 2. That said, we would like to note that while such a transform can help reduce distributional distance, in practice, it cannot perfectly mitigate the distributonal gap. This is because Sim2Real discrepancies exist not only in terms of appearance, but also content. For instance, compared to Cityscapes, synthetic scenes from GTAV tend to be much less populated with vehicle and human instances. Similarly, synthetic source images in VisDA-C contain clear backgrounds (no surrounding context) unlike their real target counterparts.

Further, combining deep network (DNN) based style-transfer operations with state-of-the-art Sim2Real UDA methods is non-trivial since -- (1) it necessitates pre-training a translation network for every Sim2Real shift, (2) running DNN based style transfer online in conjunction with UDA is expensive and (3) doing such translations for synthetic source images offline would require quality checks for different Sim2Real shifts. Our intent behind experimenting with PASTA and RandAug as AUG transforms in AUGCAL was to consider simple eligible transforms (based on properties 1 and 2) that are inexpensive to combine with Sim2Real UDA methods while having shown demonstrable benefits in such settings (as shown in [A]).

[A] - PASTA: Proportional amplitude spectrum training augmentation for syn-to-real domain generalization, ICCV 2023.

---

### Meta-Review · Area_Chair_XGyZ · 2023-12-13

**Metareview:**

This paper receives 2x marginally above acceptance threshold and 1x accept, good paper. The strengths of this paper are: The theory is nice and understandable. Extensive experiments on different tasks, datasets and UDA backbones are provided in the paper to evaluate the proposed method. Baselines with AUGCAL outperform those without AUGCAL in all metrics. Augmentation and calibration are complementary and can be integrated easily to achieve the goal. The paper provides insights on how the calibration loss on synthetic data can reduce miscalibration on real data via a theoretical analysis. This is beneficial for other related research fields. Although there are minor weaknesses of the paper, they are adequately addressed by the authors during rebuttal.

**Justification For Why Not Higher Score:**

This paper receives 2x marginally above acceptance threshold and 1x accept, good paper, and thus it should be accepted as poster.

**Justification For Why Not Lower Score:**

The minor weaknesses are adequately addressed by the authors.

---

### Decision · Program_Chairs · 2024-01-16

Accept (poster)